# Efficient Algorithms for Generalized Linear Bandits with Heavy-tailed Rewards

**Bo Xue**[1,2], **Yimu Wang**[3], **Yuanyu Wan**[4], **Jinfeng Yi**[5], **Lijun Zhang**[6,7,*]

[1]Department of Computer Science, City University of Hong Kong, Hong Kong, China
[2]The City University of Hong Kong Shenzhen Research Institute, Shenzhen, China
[3]Cheriton School of Computer Science, University of Waterloo, Waterloo, Canada
[4]School of Software Technology, Zhejiang University, Ningbo, China
[5]JD AI Research, Beijing, China
[6]National Key Laboratory for Novel Software Technology, Nanjing University, Nanjing, China
[7]Peng Cheng Laboratory, Shenzhen, China
boxue4-c@my.cityu.edu.hk, yimu.wang@uwaterloo.ca, wanyy@zju.edu.cn
yijinfeng@jd.com, zhanglj@lamda.nju.edu.cn

## Abstract

This paper investigates the problem of generalized linear bandits with heavy-tailed rewards, whose $(1 + \epsilon)$-th moment is bounded for some $\epsilon \in (0, 1]$. Although there exist methods for generalized linear bandits, most of them focus on bounded or sub-Gaussian rewards and are not well-suited for many real-world scenarios, such as financial markets and web-advertising. To address this issue, we propose two novel algorithms based on truncation and mean of medians. These algorithms achieve an almost optimal regret bound of $\widetilde{O}(dT^{\frac{1}{1+\epsilon}})$, where $d$ is the dimension of contextual information and $T$ is the time horizon. Our truncation-based algorithm supports online learning, distinguishing it from existing truncation-based approaches. Additionally, our mean-of-medians-based algorithm requires only $O(\log T)$ rewards and one estimator per epoch, making it more practical. Moreover, our algorithms improve the regret bounds by a logarithmic factor compared to existing algorithms when $\epsilon = 1$. Numerical experimental results confirm the merits of our algorithms.

## 1   Introduction

The multi-armed bandits (MAB) is a powerful framework to model the sequential decision-making process with limited information [Robbins, 1952], which has been found applications in various areas such as medical trails [Villar *et al.*, 2015] and advertisement placement [Bubeck and Cesa-Bianchi, 2012]. In the classical $K$-armed bandit problem, an agent selects one of the $K$ arms and receives a reward drawn independently and identically distributed from an unknown distribution associated with the chosen arm. The goal of the agent is to maximize the cumulative rewards through the trade-off between exploration and exploitation, i.e., pulling the arms that may potentially give better outcomes while also exploiting the knowledge gained from previous trials to select the optimal arm.

One fundamental limitation of MAB is that it ignores contextual information in some scenarios such as advertisement placement [Lattimore and Szepesvári, 2020], where features of users and products can provide valuable guidance for decision making. In these cases, decisions should not only rely on rewards from previous epochs but also the contextual information from both past and current epochs. Stochastic Linear Bandits (SLB) has emerged as the most popular model in the last decade to address this limitation, assuming a linear relationship between the contextual vector and the expected reward

---

[*]Lijun Zhang is the corresponding author.

37th Conference on Neural Information Processing Systems (NeurIPS 2023).

Table 1: Summary of the existing work for the linear bandits with heavy-tailed rewards. CC is the abbreviation of computational complexity.

| | Regret | CC_Truncation | CC_MoM | Arms | Model |
|---|---|---|---|---|---|
| Medina and Yang [2016] | $\widetilde{O}(dT^{\frac{3}{4}})$ | $O(d^2T)$ | $O(d^2T/\log T)$ | infinite | SLB |
| Shao *et al.* [2018] | $\widetilde{O}(d\sqrt{T})$ | $O(d^3T + d^2T^2)$ | $O(d^2T \cdot \log T)$ | infinite | SLB |
| Xue *et al.* [2020] | $\widetilde{O}(\sqrt{dT})$ | $O(d^2T^2)$ | $O(d^2T)$ | finite | SLB |
| This work | $\widetilde{O}(d\sqrt{T})$ | $O(d^2T)$ | $O(d^2T/\log T)$ | infinite | GLB |

[Auer, 2002; Dani *et al.*, 2008; Abbasi-yadkori *et al.*, 2011; Hu *et al.*, 2021; Alieva *et al.*, 2021; Yang *et al.*, 2022; He *et al.*, 2022; Bengs *et al.*, 2022]. However, in many real-world applications, such as social network [Filippi *et al.*, 2010], the assumption of Poisson or logistic relation between the expected reward and contextual vector has demonstrated better performance, which motivates the study of generalized linear bandits (GLB). In each round, the agent first observes a decision set $\mathcal{D}_t \subset \mathbb{R}^d$ composed of contextual vectors. Then, the agent selects an arm $\boldsymbol{x}_t \in \mathcal{D}_t$ and receives a reward $y_t$ satisfying the expectation,

$$\mathrm{E}[y_t|\boldsymbol{x}_t] = \mu(\boldsymbol{x}_t^\top \boldsymbol{\theta}_*) \tag{1}$$

where $\boldsymbol{\theta}_*$ is the inherent vector and $\mu(\cdot)$ is the link function, such as the identity function or the logistic function. The performance of the agent is measured by the regret such that

$$R(T) = \sum_{t=1}^{T} \left( \mu(\tilde{\boldsymbol{x}}_t^\top \boldsymbol{\theta}_*) - \mu(\boldsymbol{x}_t^\top \boldsymbol{\theta}_*) \right)$$

where $\tilde{\boldsymbol{x}}_t = \mathrm{argmax}_{\boldsymbol{x} \in \mathcal{D}_t} \mu(\boldsymbol{x}^\top \boldsymbol{\theta}_*)$ represents the optimal arm in the set $\mathcal{D}_t$.

Extensive research has been conducted on the GLB, with most assuming sub-Gaussian rewards [Filippi *et al.*, 2010; Li *et al.*, 2012, 2017; Jun *et al.*, 2017; Lu *et al.*, 2019; Zhou *et al.*, 2019; Han *et al.*, 2021; Li and Wang, 2022]. However, it has been observed that in certain sequential decision-making scenarios, such as financial markets [Cont and Bouchaud, 2000], the occurrence of extreme returns is much more frequent than the standard normal distribution. This phenomenon is known as heavy-tailed behavior [Foss *et al.*, 2013], where the existing algorithms are not suitable. To address this limitation, in this study, we focus on the GLB with heavy-tailed rewards [Bubeck *et al.*, 2013], i.e., the reward obtained at $t$-th round satifies the condition

$$\mathrm{E}[|y_t|^{1+\epsilon}] \leq u$$

for some $\epsilon \in (0, 1]$ and $u > 0$. Different from the traditional sub-Gaussian setting, heavy-tailed rewards do not decay exponentially and the estimation of expected rewards is significantly impacted.

According to the distinguishing characteristic of heavy-tailed distributions where extreme values occur with high probability, previous studies have developed three main strategies to address the issue in parameter estimation [Audibert and Catoni, 2011; Hsu and Sabato, 2014; Zhang and Zhou, 2018; Ray Chowdhury and Gopalan, 2019; Lugosi and Mendelson, 2021; Zhong *et al.*, 2021; Huang *et al.*, 2022; Diakonikolas *et al.*, 2022; Gorbunov *et al.*, 2022; Kamath *et al.*, 2022; Li and Liu, 2022; Gou *et al.*, 2023]. One such strategy is truncation Audibert and Catoni [2011], which mitigates the impact of extreme values by truncating them. A recently proposed strategy is the mean of medians approach [Zhong *et al.*, 2021], which involves partitioning the samples drawn from the heavy-tailed distribution into multiple groups, taking the median within each group, and computing the mean of these medians. It intuitively reduces the impact of extreme samples, as extreme samples are distributed to both sides, thus the median value is more robust. The third strategy is median of means [Hsu and Sabato, 2014], which adjusts the order of calculating mean and taking the median in the mean of medians strategy.

Most existing algorithms for heavy-tailed bandit problems are derived from aforementioned three strategies, with a primary focus on the SLB model [Medina and Yang, 2016; Shao *et al.*, 2018; Xue *et al.*, 2020]. To provide a comprehensive overview and facilitate comparison, we present a summary of our results and previous findings on linear bandits with heavy-tailed rewards in Table 1. For the sake of clarity, the presented regret bounds in Table 1 are under the assumption that the rewards have finite variance. The computational complexity only takes into account multiplication and division operations. Although Shao *et al.* [2018] and Xue *et al.* [2020] achieve nearly optimal regret for

infinite-armed and finite-armed SLB, respectively, their algorithms are computationally expensive. The latest work utilizing the mean of medians approach demonstrates efficiency but is limited to symmetric rewards [Zhong *et al.*, 2021]. Therefore, designing efficient heavy-tailed algorithms for GLB with symmetric and asymmetric rewards is an interesting and non-trivial challenge.

Through the delicate employment of heavy-tailed strategies, our contributions to the generalized linear bandit problem with heavy-tailed rewards can be summeraized as follows:

- We develop two novel algorithms, CRTM and CRMM, which utilize the truncation strategy and mean of medians strategy, respectively. Both algorithms exhibit a sublinear regret bound of $\widetilde{O}(dT^{\frac{1}{1+\epsilon}})$, which is amolst optimal as the lower bound is $\Omega(dT^{\frac{1}{1+\epsilon}})$ [Shao *et al.*, 2018].
- CRTM reduces the computational complexity from $O(T^2)$ to $O(T)$ when compared to existing truncation-based algorithms [Shao *et al.*, 2018; Xue *et al.*, 2020], while CRMM reduces the number of estimator required per round from $O(\log T)$ to only one, as compared to existing median-of-means-based algorithms [Shao *et al.*, 2018; Xue *et al.*, 2020].
- When $\epsilon = 1$, the regret bounds of CRTM and CRMM improves a logarithmic factor of order $\frac{1}{2\alpha}$ and $\frac{1}{2\alpha} - \frac{1}{2}$ for some $\alpha \in (0,1)$, respectively, over the recently proposed method of Zhong *et al.* [2021][2]. Notably, CRTM extends the method of Zhong *et al.* [2021] from symmetric rewards to general case, making CRTM more practical.
- We conduct numerical experiments to demonstrate that our proposed algorithms not only achieve a lower regret bound but also require fewer computational resources when applied to heavy-tailed bandit problems.

## 2 Related Work

In this section, we briefly review the related work on linear bandits. Through out the paper, the $p$-norm of a vector $\boldsymbol{x} \in \mathbb{R}^d$ is $\|\boldsymbol{x}\|_p = (|x_1|^p + \ldots + |x_d|^p)^{1/p}$. Given a positive definite matrix $\mathbf{A} \in \mathbb{R}^{d \times d}$, the weighted Euclidean norm of the vector $\boldsymbol{x}$ is $\|\boldsymbol{x}\|_{\mathbf{A}} = \sqrt{\boldsymbol{x}^\top \mathbf{A} \boldsymbol{x}}$.

### 2.1 Generalized Linear Bandits

Filippi *et al.* [2010] was the first to address the generalized linear bandit problem and proposed an algorithm with a regret bound of $\widetilde{O}(d\sqrt{T})$. However, their algorithm is not efficient as it requires storing all the action-feedback pairs encountered so far and performing maximum likelihood estimation at each step. A notable improvement was presented by Zhang *et al.* [2016] with the introduction of an efficient algorithm called OL$^2$M, whose space and time complexity at each epoch does not grow over time and achieves a $\widetilde{O}(d\sqrt{T})$ regret. However, their algorithm is limited to the logistic link function. Later, Jun *et al.* [2017] extended OL$^2$M to generic link functions while still maintaining the $\widetilde{O}(d\sqrt{T})$ regret bound. Ding *et al.* [2021] proposed another efficient generalized linear bandit algorithm following the line of Thompson sampling scheme.

The main challenge in the bandit problem is the trade-off between exploration and exploitation. To address this issue, the most commonly used approach is the confidence-region-based method, specifically for the linear bandit model with infinite arms [Dani *et al.*, 2008; Abbasi-yadkori *et al.*, 2011; Zhang *et al.*, 2016]. Here we take the algorithm OL$^2$M to give a brife introduction to this approach [Zhang *et al.*, 2016]. With the arrival of a new trial $(\boldsymbol{x}_t, y_t)$ in the $t$-th epoch, OL$^2$M first constructs a surrogate loss $\ell_t(\boldsymbol{\theta})$ satisfying $\nabla \ell_t(\boldsymbol{\theta}) = (-y_t + \mu(\boldsymbol{x}_t^\top \boldsymbol{\theta}))\boldsymbol{x}_t$. Then, OL$^2$M employs a variant of the online Newton step (ONS) to update the estimated parameters, i.e.,

$$\hat{\boldsymbol{\theta}}_{t+1}^N = \underset{\boldsymbol{\theta} \in \mathbb{R}^d}{\operatorname{argmin}} \frac{\|\boldsymbol{\theta} - \hat{\boldsymbol{\theta}}_t^N\|_{\mathbf{V}_{t+1}}^2}{2} + \langle \boldsymbol{\theta} - \hat{\boldsymbol{\theta}}_t^N, \nabla \ell_t(\hat{\boldsymbol{\theta}}_t^N) \rangle. \tag{2}$$

Here, $\mathbf{V}_{t+1} = \mathbf{V}_t + \frac{\kappa}{2} \boldsymbol{x}_t \boldsymbol{x}_t^\top$ for $\kappa > 0$, and the initialized matrix $\mathbf{V}_1 = \lambda \mathbf{I}_d$ for $\lambda > 0$. Subsequently, OL$^2$M constructs a confidence region $\mathcal{C}_{t+1}$ centered at the estimated parameter $\hat{\boldsymbol{\theta}}_{t+1}^N$, such that

$$\mathcal{C}_{t+1} = \left\{ \boldsymbol{\theta} \in \mathbb{R}^d \big| \|\boldsymbol{\theta} - \hat{\boldsymbol{\theta}}_{t+1}^N\|_{\mathbf{V}_{t+1}}^2 \leq \gamma_{t+1} \right\} \tag{3}$$

---

[2]For $\epsilon_1 > \epsilon_2 > 0$, if the $(1 + \epsilon_1)$-th moment of rewards exists, then the $(1 + \epsilon_2)$-th moment of rewards is bounded [Xue *et al.*, 2020]. Thus, CRTM and CRMM achieve this regret improvement when $\epsilon \geq 1$.

where $\gamma_{t+1} = O(d \log t)$ indicating the uncertainty of the estimation and the unknown parameter $\boldsymbol{\theta}_*$ lies in this region with high probability. Finally, OL$^2$M selects the most promising arm $\boldsymbol{x}_{t+1}$ according to the principle of "optimization in the face of uncertainty", i.e.,

$$(\boldsymbol{x}_{t+1}, \tilde{\boldsymbol{\theta}}_{t+1}) = \underset{\boldsymbol{x} \in \mathcal{D}_{t+1}, \boldsymbol{\theta} \in \mathcal{C}_{t+1}}{\operatorname{argmax}} \langle \boldsymbol{x}, \boldsymbol{\theta} \rangle. \tag{4}$$

## 2.2 Bandit Learning with Heavy-tailed Rewards

Most of the existing work developed heavy-tailed bandit algorithms using truncation and median of means strategies [Bubeck *et al.*, 2013; Medina and Yang, 2016; Shao *et al.*, 2018; Xue *et al.*, 2020; Huang *et al.*, 2022]. Bubeck *et al.* [2013] first conducted extensive research on multi-armed bandits with heavy-tailed rewards and achieved a logarithmic regret bound. Medina and Yang [2016] extended it to the SLB model and introduced two algorithms that achieve regret bounds of $\widetilde{O}(dT^{\frac{2+\epsilon}{2(1+\epsilon)}})$ and $\widetilde{O}(d^{\frac{1}{2}}T^{\frac{1+2\epsilon}{1+3\epsilon}} + dT^{\frac{1+\epsilon}{1+3\epsilon}})$, respectively. Shao *et al.* [2018] improved upon the results of Medina and Yang [2016] by a more delicate application of heavy-tailed strategies, achieving a regret bound of $\widetilde{O}(dT^{\frac{1}{1+\epsilon}})$. Xue *et al.* [2020] investigated the case with finite arms and provided two algorithms that attained regret bounds of $\widetilde{O}(d^{\frac{1}{2}}T^{\frac{1}{1+\epsilon}})$. Recently, Zhong *et al.* [2021] proposed the mean of medians estimator for the super heavy-tailed bandit problem, but the rewards are limited to symmetric distributions. Applying this estimator to the GLB algorithm of Jun *et al.* [2017] yields a heavy-tailed GLB algorithm that achieves the regret bound of $O(d(\log T)^{\frac{1}{2\alpha} + \frac{3}{2}}T^{\frac{1}{2}})$ for some $\alpha \in (0, 1)$. To illustrate the basic idea of adopting different heavy-tailed strategies in the bandit model, we briefly describe three representative algorithms.

For the algorithm exploiting truncation strategy, we take the algorithm TOFU as an instance [Shao *et al.*, 2018]. With the trials up to round $t$, TOFU truncates the rewards $d$ times as follows,

$$\overline{Y}_t^i = \left[ y_1 \mathbb{I}_{|u_1^i(t)y_1| \le h_t}, \dots, y_t \mathbb{I}_{|u_t^i(t)y_t| \le h_t} \right], i = 1, 2, \dots, d \tag{5}$$

where $h_t = O(t^{\frac{1-\epsilon}{2(1+\epsilon)}})$ is the truncated criterion, and $u_\tau^i(t)$ denotes the element in the $i$-th row and $\tau$-th column of matrix $\widetilde{\mathbf{V}}_{t+1}^{-1/2}\mathbf{A}_t$, $\mathbf{A}_t = [\boldsymbol{x}_1, \boldsymbol{x}_2, \dots, \boldsymbol{x}_t] \in \mathbb{R}^{d \times t}$ is the matrix composed of selected arms and $\widetilde{\mathbf{V}}_{t+1} = \mathbf{A}_t \mathbf{A}_t^\top + \mathbf{I}_d$. Using these truncated rewards, TOFU conducts an estimator as $\tilde{\boldsymbol{\theta}}_{t+1} = \widetilde{\mathbf{V}}_{t+1}^{-1/2}[\boldsymbol{u}_t^1 \cdot \overline{Y}_t^1, \dots, \boldsymbol{u}_t^d \cdot \overline{Y}_t^d]$ with $\boldsymbol{u}_t^i \cdot \overline{Y}_t^i = \sum_{\tau=1}^t u_\tau^i(t)y_\tau \mathbb{I}_{|u_\tau^i y_\tau| \le h_t}$. TOFU then constructs a confidence region centered on this estimator and selects the promising arm, similar to (3) and (4). Notice that the scalarized parameters $\{u_\tau^i(t)\}_{\tau=1}^t$ are updated at each epoch, requiring TOFU to store the learning history and truncate all rewards at each epoch. Thus, TOFU is not an online method.

For the algorithm exploiting median of means strategy, it's common to play the chosen arm $r$ times and get $r$ sequences of rewards $\{Y_t^j\}_{j=1}^r$, where $Y_t^j = [y_1^j, \dots, y_t^j]$ is the $j$-th reward sequence up to epoch $t$. MENU executes least square estimation for each reward sequence and get $r$ estimators, i.e.,

$$\hat{\boldsymbol{\theta}}_{t+1}^j = \underset{\boldsymbol{\theta} \in \mathbb{R}^d}{\operatorname{argmin}} \|\mathbf{A}_t^\top \boldsymbol{\theta} - Y_t^j\|_2^2 + \|\boldsymbol{\theta}\|_2^2, \ j = 1, 2, \dots, r \tag{6}$$

where $r = O(\log T)$ [Shao *et al.*, 2018]. Then, the median of means strategy adopted by MENU is operated as follows,

$$m_j = \text{median of } \left\{ \|\hat{\boldsymbol{\theta}}_{t+1}^j - \hat{\boldsymbol{\theta}}_{t+1}^s\|_{\widetilde{\mathbf{V}}_{t+1}} : s = 1, \dots, r \right\}. \tag{7}$$

Then, MENU takes the estimator $\hat{\boldsymbol{\theta}}_{t+1}^{k_*}$ with $k_* = \operatorname{argmin}_{j \in \{1, 2, \dots, r\}} \{m_j\}$ as the center of confidence region. Finally, MENU selects the most promising arm similar to (4).

For the mean of medians method proposed by Zhong *et al.* [2021], at each epoch $t$, the agent first plays the selected arm $\bar{r}$ times, with a value of $\bar{r} = O((\log T)^{1/\alpha})$ for some $\alpha \in (0, 1)$, and then receives rewards $\{y_t^j\}_{j=1}^{\bar{r}}$ for these plays. Subsequently, the agent randomly divides the rewards into multiple groups, with each group contains $\lceil \bar{r}^\alpha \rceil$ rewards. The agent then takes the median of each group and uses the mean of these medians to update the estimator. Notice that the expectation of the median has a bias to the expected reward other than the symmetric distribution. Thus, mean of medians strategy is limited to symmetric distribution. Another point worth mentioning is that $\bar{r}$ is too

large to try sufficient different arms. For example, the agent can only play $\lceil T/\bar{r} \rceil = 100$ different arms with $T = 10^6$ and $\alpha = 0.62$, which is obviously unreasonable[3].

# 3 Algorithms

In this section, we first introduce the generalized linear bandit model and then demonstrate two novel algorithms based on truncation and mean of medians, respectively.

## 3.1 Learning Model

The formal description of the generalized linear bandit model is as follows. In each round $t$, an agent plays an arm $\boldsymbol{x}_t \in \mathcal{D}_t$ and obtains a stochastic reward $y_t$, which is generated from a generalized linear model represented by the following equation,

$$\Pr(y_t|\boldsymbol{x}_t) = \exp\left(\frac{y_t \boldsymbol{x}_t^\top \boldsymbol{\theta}_* - m(\boldsymbol{x}_t^\top \boldsymbol{\theta}_*)}{g(\tau)} + h(y_t, \tau)\right) \tag{8}$$

where $\boldsymbol{\theta}_*$ is the inherent parameters, $\tau > 0$ is a known scale parameter, and $g(\cdot)$ and $h(\cdot, \cdot)$ are normalizers [P. McCullagh, 1989]. The expectation of $y_t$ is given by

$$\mathrm{E}[y_t|\boldsymbol{x}_t] = m'(\boldsymbol{x}_t^\top \boldsymbol{\theta}_*).$$

Thus, $m'(\cdot)$ is the link function in (1), such that $\mu(\cdot) = m'(\cdot)$. The reward model can be rewritten as

$$y_t = \mu(\boldsymbol{x}_t^\top \boldsymbol{\theta}_*) + \eta_t$$

where $\eta_t$ is a random noise satisfying the condition

$$\mathrm{E}[\eta_t|\mathcal{G}_{t-1}] = 0. \tag{9}$$

Here, $\mathcal{G}_{t-1} \triangleq \{\boldsymbol{x}_1, y_1, \ldots, \boldsymbol{x}_{t-1}, y_{t-1}, \boldsymbol{x}_t\}$ is a $\sigma$-filtration and $\mathcal{G}_0 = \emptyset$. Following the existing work [Filippi *et al.*, 2010; Jun *et al.*, 2017; Li *et al.*, 2017], we make standard assumptions as follows.

**Assumption 1** *The coefficients $\boldsymbol{\theta}_*$ and contextual vectors $\boldsymbol{x}$ are bounded, such that $\|\boldsymbol{\theta}_*\|_2 \leq S$ and $\|\boldsymbol{x}\|_2 \leq 1$ for all $\boldsymbol{x} \in \mathcal{D}_t$, where $S$ is a known constant.*

**Assumption 2** *The link function $\mu(\cdot)$ is L-Lipschitz on $[-S, S]$, and continuously differentiable on $(-S, S)$. Moreover, there exists some $\kappa > 0$ such that $\mu'(z) \geq \kappa$ and $|\mu(z)| \leq U$ for any $z \in (-S, S)$.*

## 3.2 Truncation

Our first algorithm is called Confidence Region with Truncated Mean (CRTM). The complete procedure is provided in Algorithm 1. Here, we consider the heavy-tailed setting, i.e., there exists a constant $u > 0$, the rewards admit

$$\mathrm{E}\left[|y_t|^{1+\epsilon}|\mathcal{G}_{t-1}\right] \leq u. \tag{10}$$

As we have mentioned earlier in Section 2.1, to design effective algorithms for GLB model, constructing a narrow confidence region for the underlying coefficients $\boldsymbol{\theta}_*$ is necessary. However, heavy-tailed rewards that satisfy (10) produce extreme values with high probability, resulting in a confidence region with a large radius. Therefore, a straightforward approach to settle this problem is to truncate the extreme reward to reduce its impact.

In each round $t$, CRTM first plays an arm $\boldsymbol{x}_t \in \mathcal{D}_t$ and observes the corresponding reward $y_t$. Then, CRTM truncates the reward $y_t$ using a uniform criterion $\Gamma = \widetilde{O}(T^{\frac{1-\epsilon}{2(1+\epsilon)}})$, such that

$$\tilde{y}_t = y_t \mathbb{I}_{\|\boldsymbol{x}_t\|_{\mathbf{V}_t^{-1}} |y_t| \leq \Gamma}$$

---

[3]$\alpha = 0.62$ is nearly optimal for $\epsilon = 1$ according to the experiments of Zhong *et al.* [2021].

---

**Algorithm 1** Confidence Region with Truncated Mean (CRTM)

---

**Input**: $\delta, \epsilon, u, \kappa, S, \lambda = \max\{1, \kappa/2\}$ and $T \in \mathbb{N}_+$

1: Initialize $\hat{\boldsymbol{\theta}}_1 = \mathbf{0}$ and $\mathbf{V}_1 = \lambda \mathbf{I}_d$

2: Define the truncation criterion $\Gamma = 2 \left( u / \ln(4T/\delta) \right)^{\frac{1}{1+\epsilon}} \left( d \ln \left( 1 + \frac{\kappa T}{2\lambda d} \right) / \kappa \right)^{\frac{1}{2}} T^{\frac{1-\epsilon}{2(1+\epsilon)}}$

3: **for** $t = 1, 2, \ldots, T$ **do**

4: $\quad (\boldsymbol{x}_t, \tilde{\boldsymbol{\theta}}_t) = \mathrm{argmax}_{\boldsymbol{x} \in \mathcal{D}_t, \boldsymbol{\theta} \in \mathcal{C}_t} \langle \boldsymbol{x}, \boldsymbol{\theta} \rangle$

5: $\quad$ Play the arm $\boldsymbol{x}_t$ and observe the payoff $y_t$

6: $\quad$ Truncate the observed payoff $\tilde{y}_t = y_t \mathbb{I}_{\|\boldsymbol{x}_t\|_{\mathbf{V}_t^{-1}} |y_t| \leq \Gamma}$

7: $\quad$ Compute the gradient $\nabla \tilde{\ell}_t(\hat{\boldsymbol{\theta}}_t) = (-\tilde{y}_t + \mu(\boldsymbol{x}_t^\top \hat{\boldsymbol{\theta}}_t)) \boldsymbol{x}_t$

8: $\quad$ Update $\mathbf{V}_{t+1} = \mathbf{V}_t + \frac{\kappa}{2} \boldsymbol{x}_t \boldsymbol{x}_t^\top$

9: $\quad$ Update the estimator

$$\hat{\boldsymbol{\theta}}_{t+1} = \mathrm{argmin}_{\|\boldsymbol{\theta}\|_2 \leq S} \frac{\|\boldsymbol{\theta} - \hat{\boldsymbol{\theta}}_t\|_{\mathbf{V}_{t+1}}^2}{2} + \langle \boldsymbol{\theta} - \hat{\boldsymbol{\theta}}_t, \nabla \tilde{\ell}_t(\hat{\boldsymbol{\theta}}_t) \rangle$$

10: $\quad$ Construct the confidence region

$$\mathcal{C}_{t+1} = \left\{ \boldsymbol{\theta} \in \mathbb{R}^d \big| \|\boldsymbol{\theta} - \hat{\boldsymbol{\theta}}_{t+1}\|_{\mathbf{V}_{t+1}}^2 \leq \gamma \right\}$$

11: **end for**

---

where $\mathbf{V}_t = \mathbf{V}_{t-1} + \frac{\kappa}{2} \boldsymbol{x}_{t-1} \boldsymbol{x}_{t-1}^\top$ with $\mathbf{V}_1 = \lambda \mathbf{I}_d$. Here, $\kappa$ is defined in Assumption 2 and $\lambda = \max\{1, \kappa/2\}$. With the processed action-reward pair $(\boldsymbol{x}_t, \tilde{y}_t)$, CRTM computes the gradient of the loss function as

$$\nabla \tilde{\ell}_t(\boldsymbol{\theta}) = (-\tilde{y}_t + \mu(\boldsymbol{x}_t^\top \boldsymbol{\theta})) \boldsymbol{x}_t, \tag{11}$$

where $\tilde{\ell}_t(\cdot)$ is the negative log-likelihood of the generalized linear model (8). After that, CRTM employs a variant of online Newton step (ONS) to update its estimator, given by

$$\hat{\boldsymbol{\theta}}_{t+1} = \mathrm{argmin}_{\|\boldsymbol{\theta}\|_2 \leq S} \frac{\|\boldsymbol{\theta} - \hat{\boldsymbol{\theta}}_t\|_{\mathbf{V}_{t+1}}^2}{2} + \langle \boldsymbol{\theta} - \hat{\boldsymbol{\theta}}_t, \nabla \tilde{\ell}_t(\hat{\boldsymbol{\theta}}_t) \rangle.$$

Equipped with above estimation, CRTM constructs the confidence region $\mathcal{C}_{t+1}$ where the inherent parameters $\boldsymbol{\theta}_*$ lies in with high probability, such that

$$\mathcal{C}_{t+1} = \left\{ \boldsymbol{\theta} \in \mathbb{R}^d \big| \|\boldsymbol{\theta} - \hat{\boldsymbol{\theta}}_{t+1}\|_{\mathbf{V}_{t+1}}^2 \leq \gamma \right\}$$

where $\gamma = \widetilde{O}(T^{\frac{1-\epsilon}{1+\epsilon}})$ denotes the width of the confidence region, and details are shown in Theorem 1. Given the confidence region $\mathcal{C}_{t+1}$, the most promising arm $\boldsymbol{x}_{t+1}$ can be obtained through the following maximize operation,

$$(\boldsymbol{x}_{t+1}, \tilde{\boldsymbol{\theta}}_{t+1}) = \mathrm{argmax}_{\boldsymbol{x} \in \mathcal{D}_{t+1}, \boldsymbol{\theta} \in \mathcal{C}_{t+1}} \langle \boldsymbol{x}, \boldsymbol{\theta} \rangle$$

since $\mu(\cdot)$ is monotonically increasing according to Assumption 2.

Although there exists several heavy-tailed linear bandit algorithms based on the truncation strategy, such as TOFU [Shao *et al.*, 2018] and BTC [Xue *et al.*, 2020], CRTM differs from them in two aspects. Firstly, both TOFU and BTC have to store the historical rewards and truncate them at each epoch, resulting in a computational complexity of $O(T^2)$. In contrast, CRTM achieves online learning by processing only the reward of current round, whose computational complexity is $O(T)$. Secondly, while TOFU and BTC are designed for SLB model and calculate the estimator via least-squares estimation, CRTM is designed for the GLB model and updates the estimator using the ONS method, which makes the analytical techniques fundamentally different. Theorem 1 provides a tight confidence region, and its proof relies on the induced method because ONS is an iteratively updated method. Due to the page limit, we provide the detailed proof in the Appendix A.2.

**Theorem 1** *If the rewards satisfy* (9) *and* (10)*, then with probability as least $1 - \delta$, the confidence region in CRTM is*

$$\|\boldsymbol{\theta} - \hat{\boldsymbol{\theta}}_{t+1}\|_{\mathbf{V}_{t+1}}^2 \leq \gamma, \forall t \geq 0$$

**Algorithm 2** Confidence Region with Mean of Medians (CRMM)

---

**Input**: $\delta, \epsilon, v, \kappa, S, \lambda = \max\{1, \kappa/2\}$ and $T \in \mathbb{N}_+$

1: Initialize $\hat{\boldsymbol{\theta}}_1 = \mathbf{0}$, $\mathbf{V}_1 = \lambda \mathbf{I}_d$ and $\gamma_1 = \lambda S^2$
2: $r = \lceil 16 \ln \frac{4T}{\delta} \rceil$ and $T_0 = \lfloor T/r \rfloor$
3: **for** $t = 1, 2, \ldots, T_0$ **do**
4:     $(\boldsymbol{x}_t, \hat{\boldsymbol{\theta}}_t) = \operatorname{argmax}_{\boldsymbol{x} \in \mathcal{D}, \boldsymbol{\theta} \in \mathcal{C}_t} \langle \boldsymbol{x}, \boldsymbol{\theta} \rangle$
5:     Play the arm $\boldsymbol{x}_t$ $r$ times and observe the rewards $\{y_t^1, y_t^2, \ldots, y_t^r\}$
6:     Take the median of $\{y_t^1, y_t^2, \ldots, y_t^r\}$ as $\bar{y}_t$
7:     Compute the gradient $\nabla \bar{\ell}_t(\hat{\boldsymbol{\theta}}_t) = (-\bar{y}_t + \mu(\boldsymbol{x}_t^\top \hat{\boldsymbol{\theta}}_t))\boldsymbol{x}_t$
8:     Update $\mathbf{V}_{t+1} = \mathbf{V}_t + \frac{\kappa}{2}\boldsymbol{x}_t\boldsymbol{x}_t^\top$
9:     Compute the center of confidence region

$$\hat{\boldsymbol{\theta}}_{t+1} = \underset{\|\boldsymbol{\theta}\|_2 \leq S}{\operatorname{argmin}} \frac{\|\boldsymbol{\theta} - \hat{\boldsymbol{\theta}}_t\|_{\mathbf{V}_{t+1}}^2}{2} + \langle \boldsymbol{\theta} - \hat{\boldsymbol{\theta}}_t, \nabla \bar{\ell}_t(\hat{\boldsymbol{\theta}}_t) \rangle$$

10:    Construct the confidence region

$$\mathcal{C}_{t+1} = \left\{ \boldsymbol{\theta} \in \mathbb{R}^d \big| \|\boldsymbol{\theta} - \hat{\boldsymbol{\theta}}_{t+1}\|_{\mathbf{V}_{t+1}}^2 \leq \gamma_{t+1} \right\}$$

11: **end for**

---

*where*

$$\gamma = 224 u^{\frac{2}{1+\epsilon}} \ln(4T/\delta)^{\frac{2\epsilon}{1+\epsilon}} T^{\frac{1-\epsilon}{1+\epsilon}} \frac{4d}{\kappa} \ln\left(1 + \frac{\kappa T}{2\lambda d}\right) + 2\lambda S^2 + \frac{48 U^2 d}{\kappa} \ln\left(1 + \frac{\kappa T}{2\lambda d}\right).$$

With above confidence region, the regret bound of CRTM is explicitly given as follows.

**Theorem 2** *If the rewards satisfy* (9) *and* (10)*, then with probability at least* $1 - \delta$*, the regret of CRTM satisfies*

$$R(T) \leq O\left(d(\log T)^{\frac{1+2\epsilon}{1+\epsilon}} T^{\frac{1}{1+\epsilon}}\right).$$

**Remark:** The above theorem establishes a $\widetilde{O}(dT^{\frac{1}{1+\epsilon}})$ regret bound with the assumption that the $(1 + \epsilon)$-th moment of the rewards is bounded for some $\epsilon \in (0, 1]$. Existing algorithms based on truncation is time-consuming because they need to store the learning history and truncate all historical rewards at each epoch [Shao *et al.*, 2018; Xue *et al.*, 2020]. Unlike the recently proposed mean of medians method which is limited in symmetric rewards [Zhong *et al.*, 2021], CRTM expands it to asymmetric and achieves an improved regret bound by a factor of $O((\log T)^{\frac{1}{2\alpha}})$ for some $\alpha \in (0, 1)$ if $\epsilon = 1$. Furthermore, CRTM is almost optimal as the lower bound is $\Omega(dT^{\frac{1}{1+\epsilon}})$ [Shao *et al.*, 2018].

### 3.3 Mean of Medians

In this section, we present our second algorithm, referred to as Confidence Region with Mean of Medians (CRMM), which shares a similar framework with CRTM but uses a different mean of medians estimator. The complete procedure is outlined in Algorithm 2. CRMM requires that for some $\epsilon \in (0, 1]$, the $1 + \epsilon$ central moment of the rewards is bounded, and the distribution of rewards is symmetric. Precisely, for some $\epsilon \in (0, 1]$, there exists a constant $v > 0$ such that the rewards satisfy

$$\mathrm{E}\left[|\eta_t|^{1+\epsilon} | \mathcal{G}_{t-1}\right] \leq v \text{ and } p(\eta_t) = p(-\eta_t). \tag{12}$$

At each epoch $t$, CRMM plays the selected arm $\boldsymbol{x}_t$ $r$ times, generating rewards $\{y_t^1, \ldots, y_t^r\}$ with $r = O(\log T)$. To obtain a robust estimation using mean of medians strategy, CRMM first takes the median of $\{y_t^1, \ldots, y_t^r\}$, denoted by $\bar{y}_t$. Subsequently, CRMM computes the gradient with the arm-reward pair $(\boldsymbol{x}_t, \bar{y}_t)$ through the operation similar to (11). Then, CRMM employs a variant of ONS to update the estimator and construct the confidence region $\mathcal{C}_{t+1}$ centered on the new estimator. The details about the constructed confidence region is given in Theorem 3.

Compared to existing bandit algorithms that utilize the median of means strategy, the primary difference lies in the item chosen as the "means". As we have introduced in (7), MENU of Shao *et al.* [2018] uses the distance between different estimators as the "means". BMM of Xue *et al.* [2020] calculates multiple estimated rewards for each arm and treats them as the "means". Both MENU and BMM require $O(\log T)$ estimators during each round, whereas CRMM only requires one estimator. Moreover, compared to the mean of medians approach [Zhong *et al.*, 2021], CRMM plays each selected arm fewer times, leading to more model updates, which is critical based on experimental results. Since the chosen arm has to be played multiple times, we assume the arm set for CRMM is static, such that $\mathcal{D}_t = \mathcal{D}$ for $t > 0$, which is a common assumption [Medina and Yang, 2016; Zhang *et al.*, 2016; Lu *et al.*, 2019]. The following theorem guarantees a tight confidence region.

**Theorem 3** *If the rewards satisfy* (9) *and* (12)*, then with probability as least* $1 - 2\delta$*, the confidence region in CRMM is*

$$\|\boldsymbol{\theta} - \hat{\boldsymbol{\theta}}_{t+1}\|^2_{\mathbf{V}_{t+1}} \leq \gamma_{t+1}, \forall t \geq 0$$

*where*

$$\gamma_{t+1} = \left(4U^2 + C\rho t^{\frac{1-\epsilon}{1+\epsilon}}\right)\frac{4d}{\kappa}\ln\left(1 + \frac{\kappa t}{2\lambda d}\right) + \lambda S^2 + \frac{2\rho^2}{\kappa}t^{\frac{1-\epsilon}{1+\epsilon}},$$

$$\rho = 2C\ln(4T/\delta) + 2C^{-\epsilon}rv, \ C = (4v)^{\frac{1}{1+\epsilon}}.$$

With above confidence region, we prove the following regret bound for CRMM.

**Theorem 4** *If the rewards satisfy* (9) *and* (12)*, then with probability at least* $1 - 2\delta$*, the regret of CRMM satisfies*

$$R(T) \leq O\left(d(\log T)^{\frac{3}{2} + \frac{\epsilon}{1+\epsilon}}T^{\frac{1}{1+\epsilon}}\right).$$

**Remark:** Theorem 3 clarifies that if the rewards have a finite $1 + \epsilon$ central moment for some $\epsilon \in (0, 1]$, CRMM achieves a regret bound of $\widetilde{O}(dT^{\frac{1}{1+\epsilon}})$. This bound reduces to $\widetilde{O}(d\sqrt{T})$ when $\epsilon = 1$, indicating that CRMM achieves the same order as the bounded rewards assumption regarding both $d$ and $T$ [Zhang *et al.*, 2016; Jun *et al.*, 2017]. Compared to the approach of Zhong *et al.* [2021], CRMM enhances the bound by an order of $O((\log T)^{\frac{1}{2\alpha} - \frac{1}{2}})$ for a fixed $\alpha \in (0, 1)$ if $\epsilon = 1$.

# 4 Experiments

This section demonstrates the improvement of our algorithms by numerical experiments. Firstly, we show the effectiveness of our algorithms in dealing with heavy-tailed problems by comparing their regret to that of existing generalized linear bandit algorithms. Secondly, we evaluate the efficiency of our algorithms by comparing their time consumption to other existing algorithms designed for heavy-tailed bandit problems. All algorithms are implemented using PyCharm 2022 and tested on a laptop with a 2.5GHz CPU and 32GB of memory.

## 4.1 Regret Comparison

To assess the enhancement of our algorithms in handling heavy-tailed problems, we utilize the vanilla GLB algorithms, specifically OL$^2$M [Zhang *et al.*, 2016] and GLOC [Jun *et al.*, 2017], as baselines. Additionally, we incorporate the mean of medians method proposed by Zhong *et al.* [2021] into OL$^2$M and GLOC, resulting in another two baselines OL$^2$M_mom and GLOC_mom, respectively. All algorithms are configured with $\epsilon = 1$, $\delta = 0.01$, and $T = 10^6$.

Let $\boldsymbol{\theta}_* = \mathbf{1}/\sqrt{d} \in \mathbb{R}^d$, where $\mathbf{1}$ is an all-1 vector and $\|\boldsymbol{\theta}_*\|_2 = 1$. The number of arms is set to $K = 20$, and the feature dimension is $d = 10$. Each component of the contextual vector $\boldsymbol{x}_t$ is uniformly sampled from the interval $[0, 1]$, and then normalized to be unit norm, i.e., $\|\boldsymbol{x}_t\|_2 = 1$. We tune the width of the confidence region following the common practice in bandit learning [Zhang *et al.*, 2016; Jun *et al.*, 2017]. Precisely, with $c$ being a tuning parameter searched within $[1e^{-4}, 1]$, the width of the confidence region for OL$^2$M and GLOC are set as $\gamma_t = cd\ln(t/\lambda + 1)$ and $\gamma_t = c\sum_{\tau=1}^{t}(\mu(\boldsymbol{x}_\tau^\top \hat{\boldsymbol{\theta}}_\tau) - y_\tau)^2\|\boldsymbol{x}_\tau\|^2_{\mathbf{V}_\tau^{-1}}$, respectively. In addition, the radius of the confidence region is set as $cd\ln(4T/\delta)^{\frac{2\epsilon}{1+\epsilon}}\ln(T/(d\lambda) + 1)T^{\frac{1-\epsilon}{1+\epsilon}}$ for CRTM, and $cd\ln(t/(d\lambda) + 1)t^{\frac{1-\epsilon}{1+\epsilon}}$ for

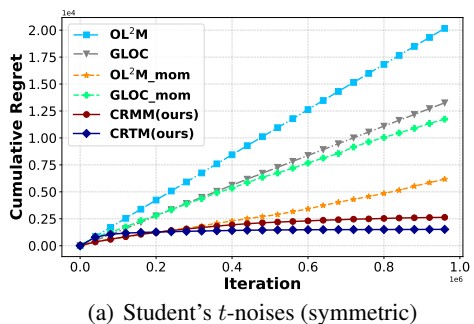

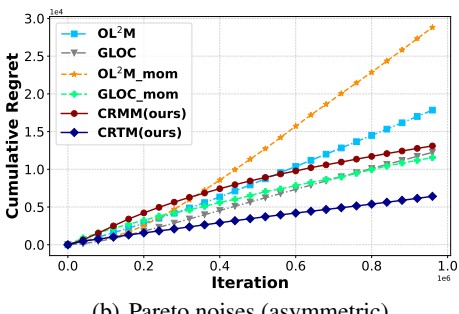

| (a) Student's $t$-noises (symmetric) | (b) Pareto noises (asymmetric) |

Figure 1: Regret comparison

CRMM. For OL$^2$M_mom and GLOC_mom, the chosen arm is played $\bar{r} = (16 \ln(2T/\delta))^{1/\alpha}$ times per round, and $\alpha = 0.62$ is close to optimal according to the experiments of Zhong *et al.* [2021].

We run 10 repetitions for each algorithm and display the average regret with time evolution. According to the generalized linear bandit model, the observed reward $y_t$ is given by

$$y_t = \mu(\boldsymbol{x}_t^\top \boldsymbol{\theta}_*) + \eta_t$$

where $\mu(x) = \frac{1}{1+e^{-x}}$ is the logit model and $\eta_t$ is the heavy-tailed noises. To evaluate the algorithms performance under both symmetric and asymmetric rewards, $\eta_t$ fits the following two distributions,

(i) Student's $t$-Noise: $\eta_t \sim \frac{G(2)}{\sqrt{3\pi}G(1.5)} \left(1 + \frac{x^2}{3}\right)^{-2}$ where $G(\cdot)$ is the Gamma function;

(ii) Pareto Noise: $\eta_t \sim \frac{sx_m^s}{x^{s+1}}\mathbb{I}_{x \geq x_m}$ where $s = 3$ and $x_m = 0.01$.

Fig. 1 compares our algorithms against two vanilla GLB algorithms (OL$^2$M and GLOC), as well as these two algorithms exploiting mean of medians estimators (OL$^2$M_mom and GLOC_mom). Fig. 1(a) shows that CRTM and CRMM outperform the other four algorithms. CRTM provides the best performance, which is consistent with the theoretical guarantees. OL$^2$M_mom and GLOC_mom appear ineffective at handling heavy-tailed problems, because they update estimator only 100 times with the chosen arm played $\bar{r}$ times [Zhong *et al.*, 2021]. Fig. 1(b) presents the cumulative regrets under asymmetric noises, with CRTM still having the lowest regret curve, demonstrating its generality and effectiveness in handling heavy-tailed bandit problems. On the other hand, CRMM, GLOC_mom, and OL$^2$M_mom performs poorly in Fig. 1(b), as they can not deal with the asymmetric rewards.

### 4.2 Runtime Comparison

To demonstrate the efficiency improvement of our algorithms, we compare them with existing heavy-tailed bandit algorithms such as CRT and MoM [Medina and Yang, 2016], TOFU and MENU [Shao *et al.*, 2018], and SupBTC and SupBMM [Xue *et al.*, 2020]. Among them, CRT, TOFU and SupBTC employ truncation strategy, while MoM, MENU and SupBMM utilize the median of means strategy.

The experimental settings are the same as described in Regret Comparison section, except for the time horizon and feature dimension. We use a smaller time horizon $T = 10^4$ since TOFU is time-consuming. The feature dimension is increased to $d = 100$ to highlight the difference between SupBTC and TOFU. The computational runtimes are shown in Table 2.

Table 2: Runtime comparsion

| Algorithm | Time(s) | Algorithm | Time(s) |
|-----------|---------|-----------|---------|
| CRT | 3.1737 | MoM | 0.0630 |
| TOFU | 3931.9963 | MENU | 24.1990 |
| SupBTC | 1187.1863 | SupBMM | 0.0685 |
| CRTM | **2.2909** | CRMM | **0.0514** |

For the truncation-based algorithms, CRTM consumes the least time, while TOFU and SupBTC takes over a hundred times longer to execute than CRTM, representing a significant improvement. CRT takes only slightly longer than CRTM as both algorithms update the model online, but the regret bound of CRT is $\widetilde{O}(dT^{\frac{3}{4}})$, which is $\widetilde{O}(T^{\frac{1}{4}})$ worse

than the bound of CRTM. For median of means algorithms, CRMM has the shortest runtime. MENU takes significantly longer than the other algorithms because MENU needs to calculate the distance between $O(\log T)$ estimators.

## 5  Conclusion and Future Work

We present two algorithms, CRTM and CRMM, for the generalized linear bandit model with heavy-tailed rewards, which utilize the truncation and mean of medians strategies, respectively. Both algorithms achieve the regret bound of $\widetilde{O}(dT^{\frac{1}{1+\epsilon}})$ conditioned on a bounded $(1+\epsilon)$-th moment of rewards, where $\epsilon \in (0, 1]$. This bound is almost optimal since the lower bound of the stochastic liear bandit problem is $\Omega(dT^{\frac{1}{1+\epsilon}})$ [Shao *et al.*, 2018]. CRTM is the first truncation-based online algorithm for the heavy-tailed bandit problem that handles both symmetric and asymmetric rewards and approaches the optimal regret bound. CRMM enhances the regret bound of the the most related work by a logarithmic factor [Zhong *et al.*, 2021]. However, CRMM is limited to symmetric rewards and we will investigate to overcome this restriction in the future.

## Acknowledgments and Disclosure of Funding

This work was partially supported by the National Key R&D Program of China (2022ZD0114801), the Key Basic Research Foundation of Shenzhen (JCYJ20220818100005011), NSFC (62122037, 61921006) and the major key project of PCL (PCL2021A12).

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
