# Supplementary Material: Efficient Algorithms for Generalized Linear Bandits with Heavy-tailed Rewards

## A  Proof of Theorem 1

To ensure clarity of expression, we have divided the proof of Theorem 1 into two subsections. The first subsection establishes a general upper bound for the confidence region constructed by ONS. Building upon the first subsection, we employ the truncated technique in the second subsection to deduce the confidence region for CRTM.

### A.1  General Upper Bound of ONS

For the sake of representation, we define the loss function for the action-reward pair $(\boldsymbol{x}_t, y_t)$ as

$$\ell_t(\boldsymbol{\theta}) = -y_t \boldsymbol{x}_t^\top \boldsymbol{\theta} + m(\boldsymbol{x}_t^\top \boldsymbol{\theta}),$$

and the conditional expectation for this loss function is denoted as $f_t(\boldsymbol{\theta}) = \mathrm{E}[\ell_t(\boldsymbol{\theta})|\mathcal{G}_{t-1}]$.

First, we propose the following lemma to display the strong convexity of the loss function.

**Lemma 1** *For any $\boldsymbol{\theta}_1, \boldsymbol{\theta}_2 \in \mathbb{R}^d$ satisfying $\|\boldsymbol{\theta}_1\|_2 \leq S, \|\boldsymbol{\theta}_2\|_2 \leq S$, the inequality*

$$\ell_t(\boldsymbol{\theta}_1) - \ell_t(\boldsymbol{\theta}_2) \geq \nabla\ell_t(\boldsymbol{\theta}_2)^\top(\boldsymbol{\theta}_1 - \boldsymbol{\theta}_2) + \frac{\kappa}{2}\left(\boldsymbol{x}_t^\top\boldsymbol{\theta}_1 - \boldsymbol{x}_t^\top\boldsymbol{\theta}_2\right)^2$$

*is true for all $t > 0$.*

**Proof.** Let $L_t(z) = -y_t z + m(z), z \in [-S, S]$, then $L_t''(z) = \mu'(z) \geq \kappa$ due to Assumption 2. Thus, $L_t(z)$ is a $\kappa$-strongly convex function, which indicates that

$$L_t(z_1) - L_t(z_2) \geq L_t'(z_2)(z_1 - z_2) + \frac{\kappa}{2}\left(z_1 - z_2\right)^2.$$

Let $z_1 = \boldsymbol{x}_t^\top\boldsymbol{\theta}_1$ and $z_2 = \boldsymbol{x}_t^\top\boldsymbol{\theta}_2$, we get that

$$L_t(\boldsymbol{x}_t^\top\boldsymbol{\theta}_1) - L_t(\boldsymbol{x}_t^\top\boldsymbol{\theta}_2) \geq L_t'(\boldsymbol{x}_t^\top\boldsymbol{\theta}_2)(\boldsymbol{x}_t^\top\boldsymbol{\theta}_1 - \boldsymbol{x}_t^\top\boldsymbol{\theta}_2) + \frac{\kappa}{2}\left(\boldsymbol{x}_t^\top\boldsymbol{\theta}_1 - \boldsymbol{x}_t^\top\boldsymbol{\theta}_2\right)^2.$$

Taking $L_t(\boldsymbol{x}_t^\top\boldsymbol{\theta}) = \ell_t(\boldsymbol{\theta})$ and $L_t'(\boldsymbol{x}_t^\top\boldsymbol{\theta})\boldsymbol{x}_t = \nabla\ell_t(\boldsymbol{\theta})$ into above equation finishes the proof.  $\square$

Then, we propose Lemma 2 to show that $\boldsymbol{\theta}_*$ is the minimum point of the expected loss function.

**Lemma 2** *Suppose $\boldsymbol{\theta} \in \mathbb{R}^d$ satisfies $\|\boldsymbol{\theta}\|_2 \leq S$, then $f_t(\boldsymbol{\theta}) - f_t(\boldsymbol{\theta}_*) \geq 0$ for all $t > 0$.*

**Proof.** Recall that GLB model satisfys $\mathrm{E}[y_t|\boldsymbol{x}_t] = m'(\boldsymbol{x}_t^\top\boldsymbol{\theta}_*)$ and $\mu(\cdot) = m'(\cdot)$, thus

$$
\begin{aligned}
f_t(\boldsymbol{\theta}) - f_t(\boldsymbol{\theta}_*) &= \mathrm{E}[\ell_t(\boldsymbol{\theta}) - \ell_t(\boldsymbol{\theta}_*)|\mathcal{G}_{t-1}] \\
&= m(\boldsymbol{x}_t^\top\boldsymbol{\theta}) - m(\boldsymbol{x}_t^\top\boldsymbol{\theta}_*) - \mu(\boldsymbol{x}_t^\top\boldsymbol{\theta}_*)(\boldsymbol{x}_t^\top\boldsymbol{\theta} - \boldsymbol{x}_t^\top\boldsymbol{\theta}_*) \\
&\geq m'(\boldsymbol{x}_t^\top\boldsymbol{\theta}_*)(\boldsymbol{x}_t^\top\boldsymbol{\theta} - \boldsymbol{x}_t^\top\boldsymbol{\theta}_*) - \mu(\boldsymbol{x}_t\boldsymbol{\theta}_*)(\boldsymbol{x}_y^\top\boldsymbol{\theta} - \boldsymbol{x}_t^\top\boldsymbol{\theta}_*) \\
&= 0
\end{aligned}
$$

where the inequality holds because $m(\cdot)$ is $\kappa$-strongly convex.  $\square$

To exploit the property of ONS, we adopt the following lemma from Zhang *et al.* [2016].

**Lemma 3** *For any $t > 0$, the inequality*

$$\nabla\ell_t(\hat{\boldsymbol{\theta}}_t)^\top(\hat{\boldsymbol{\theta}}_t - \boldsymbol{\theta}_*) - \frac{1}{2}\|\nabla\ell_t(\hat{\boldsymbol{\theta}}_t)\|^2_{\mathbf{V}_{t+1}^{-1}} \leq \frac{1}{2}\left(\|\hat{\boldsymbol{\theta}}_t - \boldsymbol{\theta}_*\|^2_{\mathbf{V}_{t+1}} - \|\hat{\boldsymbol{\theta}}_{t+1} - \boldsymbol{\theta}_*\|^2_{\mathbf{V}_{t+1}}\right)$$

*holds.*

With above three lemmas, we are ready to bound the confidence region of the ONS estimation. Lemma 1 tells that

$$\ell_t(\hat{\boldsymbol{\theta}}_t) - \ell_t(\boldsymbol{\theta}_*) \leq \nabla \ell_t(\hat{\boldsymbol{\theta}}_t)^\top (\hat{\boldsymbol{\theta}}_t - \boldsymbol{\theta}_*) - \frac{\kappa}{2} \left( \boldsymbol{x}_t^\top \hat{\boldsymbol{\theta}}_t - \boldsymbol{x}_t^\top \boldsymbol{\theta}_* \right)^2 .$$

If we take expectation in both sides, it becomes

$$f_t(\hat{\boldsymbol{\theta}}_t) - f_t(\boldsymbol{\theta}_*) \leq \nabla f_t(\hat{\boldsymbol{\theta}}_t)^\top (\hat{\boldsymbol{\theta}}_t - \boldsymbol{\theta}_*) - \frac{\kappa}{2} \left( \boldsymbol{x}_t^\top \hat{\boldsymbol{\theta}}_t - \boldsymbol{x}_t^\top \boldsymbol{\theta}_* \right)^2 .$$

Lemma 2 tells that

$$
\begin{aligned}
0 \leq{}& \nabla f_t(\hat{\boldsymbol{\theta}}_t)^\top \left( \hat{\boldsymbol{\theta}}_t - \boldsymbol{\theta}_* \right) - \frac{\kappa}{2} \left( \boldsymbol{x}_t^\top \hat{\boldsymbol{\theta}}_t - \boldsymbol{x}_t^\top \boldsymbol{\theta}_* \right)^2 \\
={}& \left( \nabla f_t(\hat{\boldsymbol{\theta}}_t) - \nabla \ell_t(\hat{\boldsymbol{\theta}}_t) \right)^\top \left( \hat{\boldsymbol{\theta}}_t - \boldsymbol{\theta}_* \right) - \frac{\kappa}{2} \left( \boldsymbol{x}_t^\top \hat{\boldsymbol{\theta}}_t - \boldsymbol{x}_t^\top \boldsymbol{\theta}_* \right)^2 + \nabla \ell_t(\hat{\boldsymbol{\theta}}_t)^\top (\hat{\boldsymbol{\theta}}_t - \boldsymbol{\theta}_*).
\end{aligned}
\tag{13}
$$

According to Lemma 3, we can relax the last term in the right side of (13) and get

$$
\begin{aligned}
0 \leq{}& \left( \nabla f_t(\hat{\boldsymbol{\theta}}_t) - \nabla \ell_t(\hat{\boldsymbol{\theta}}_t) \right)^\top \left( \hat{\boldsymbol{\theta}}_t - \boldsymbol{\theta}_* \right) - \frac{\kappa}{2} \left( \boldsymbol{x}_t^\top \hat{\boldsymbol{\theta}}_t - \boldsymbol{x}_t^\top \boldsymbol{\theta}_* \right)^2 \\
& + \frac{1}{2} \|\nabla \ell_t(\hat{\boldsymbol{\theta}}_t)\|_{\mathbf{V}_{t+1}^{-1}}^2 + \frac{1}{2} \left( \|\hat{\boldsymbol{\theta}}_t - \boldsymbol{\theta}_*\|_{\mathbf{V}_{t+1}}^2 - \|\hat{\boldsymbol{\theta}}_{t+1} - \boldsymbol{\theta}_*\|_{\mathbf{V}_{t+1}}^2 \right).
\end{aligned}
\tag{14}
$$

Then, taking the gradient

$$\nabla \ell_t(\hat{\boldsymbol{\theta}}_t) = -y_t \boldsymbol{x}_t + \mu(\boldsymbol{x}_t^\top \hat{\boldsymbol{\theta}}_t) \boldsymbol{x}_t, \ \nabla f_t(\hat{\boldsymbol{\theta}}_t) = -\mu(\boldsymbol{x}_t^\top \boldsymbol{\theta}_*) \boldsymbol{x}_t + \mu(\boldsymbol{x}_t^\top \hat{\boldsymbol{\theta}}_t) \boldsymbol{x}_t$$

into inequality (14), we get that

$$
\begin{aligned}
0 \leq{}& \frac{1}{2} \left( \|\hat{\boldsymbol{\theta}}_t - \boldsymbol{\theta}_*\|_{\mathbf{V}_{t+1}}^2 - \|\hat{\boldsymbol{\theta}}_{t+1} - \boldsymbol{\theta}_*\|_{\mathbf{V}_{t+1}}^2 \right) - \frac{\kappa}{2} \left( \boldsymbol{x}_t^\top \hat{\boldsymbol{\theta}}_t - \boldsymbol{x}_t^\top \boldsymbol{\theta}_* \right)^2 \\
& + \left( y_t - \mu(\boldsymbol{x}_t^\top \boldsymbol{\theta}_*) \right) \boldsymbol{x}_t^\top (\hat{\boldsymbol{\theta}}_t - \boldsymbol{\theta}_*) + \frac{1}{2} \|(-y_t + \mu(\boldsymbol{x}_t^\top \hat{\boldsymbol{\theta}}_t)) \boldsymbol{x}_t\|_{\mathbf{V}_{t+1}^{-1}}^2 .
\end{aligned}
$$

A simple application of triangle inequality tells that

$$
\begin{aligned}
0 \leq{}& \frac{1}{2} \left( \|\hat{\boldsymbol{\theta}}_t - \boldsymbol{\theta}_*\|_{\mathbf{V}_{t+1}}^2 - \|\hat{\boldsymbol{\theta}}_{t+1} - \boldsymbol{\theta}_*\|_{\mathbf{V}_{t+1}}^2 \right) - \frac{\kappa}{2} \left( \boldsymbol{x}_t^\top \hat{\boldsymbol{\theta}}_t - \boldsymbol{x}_t^\top \boldsymbol{\theta}_* \right)^2 \\
& + \left( y_t - \mu(\boldsymbol{x}_t^\top \boldsymbol{\theta}_*) \right) \boldsymbol{x}_t^\top (\hat{\boldsymbol{\theta}}_t - \boldsymbol{\theta}_*) \\
& + \frac{1}{2} (y_t - \mu(\boldsymbol{x}_t^\top \boldsymbol{\theta}_*))^2 \|\boldsymbol{x}_t\|_{\mathbf{V}_{t+1}^{-1}}^2 + \frac{1}{2} (\mu(\boldsymbol{x}_t^\top \boldsymbol{\theta}_*) - \mu(\boldsymbol{x}_t^\top \hat{\boldsymbol{\theta}}_t))^2 \|\boldsymbol{x}_t\|_{\mathbf{V}_{t+1}^{-1}}^2 \\
\leq{}& \frac{1}{2} \left( \|\hat{\boldsymbol{\theta}}_t - \boldsymbol{\theta}_*\|_{\mathbf{V}_t}^2 - \|\hat{\boldsymbol{\theta}}_{t+1} - \boldsymbol{\theta}_*\|_{\mathbf{V}_{t+1}}^2 \right) - \frac{\kappa}{2} \left( \boldsymbol{x}_t^\top \hat{\boldsymbol{\theta}}_t - \boldsymbol{x}_t^\top \boldsymbol{\theta}_* \right)^2 \\
& + \left( y_t - \mu(\boldsymbol{x}_t^\top \boldsymbol{\theta}_*) \right) \boldsymbol{x}_t^\top (\hat{\boldsymbol{\theta}}_t - \boldsymbol{\theta}_*) \\
& + \frac{1}{2} (y_t - \mu(\boldsymbol{x}_t^\top \boldsymbol{\theta}_*))^2 \|\boldsymbol{x}_t\|_{\mathbf{V}_t^{-1}}^2 + \frac{1}{2} (\mu(\boldsymbol{x}_t^\top \boldsymbol{\theta}_*) - \mu(\boldsymbol{x}_t^\top \hat{\boldsymbol{\theta}}_t))^2 \|\boldsymbol{x}_t\|_{\mathbf{V}_t^{-1}}^2
\end{aligned}
$$

where the second equality holds because $\mathbf{V}_{t+1} = \mathbf{V}_t + \frac{\kappa}{2} \boldsymbol{x}_t \boldsymbol{x}_t^\top$. By summing the above inequality from 1 to $t$ and rearranging, the confidence region can be bounded as

$$
\begin{aligned}
& \|\hat{\boldsymbol{\theta}}_{t+1} - \boldsymbol{\theta}_*\|_{\mathbf{V}_{t+1}}^2 \\
& \leq \|\hat{\boldsymbol{\theta}}_1 - \boldsymbol{\theta}_*\|_{\mathbf{V}_1}^2 - \frac{\kappa}{2} \sum_{\tau=1}^t \left( \boldsymbol{x}_\tau^\top \hat{\boldsymbol{\theta}}_\tau - \boldsymbol{x}_\tau^\top \boldsymbol{\theta}_* \right)^2 + \sum_{\tau=1}^t \left( \mu(\boldsymbol{x}_\tau^\top \boldsymbol{\theta}_*) - \mu(\boldsymbol{x}_\tau^\top \hat{\boldsymbol{\theta}}_\tau) \right)^2 \|\boldsymbol{x}_\tau\|_{\mathbf{V}_\tau^{-1}}^2 \\
& + \sum_{\tau=1}^t 2 \left( y_\tau - \mu(\boldsymbol{x}_\tau^\top \boldsymbol{\theta}_*) \right) \boldsymbol{x}_\tau^\top (\hat{\boldsymbol{\theta}}_\tau - \boldsymbol{\theta}_*) + \sum_{\tau=1}^t \left( y_\tau - \mu(\boldsymbol{x}_\tau^\top \boldsymbol{\theta}_*) \right)^2 \|\boldsymbol{x}_\tau\|_{\mathbf{V}_\tau^{-1}}^2 .
\end{aligned}
\tag{15}
$$

Until now, we have proven an upper bound for the ONS method updated with a general action-reward pair $(\boldsymbol{x}_t, y_t)$, and the bound is shown in equation (15).

## A.2 Truncated Upper Bound of CRTM

CRTM updates the estimator with a truncated action-reward pair $(\boldsymbol{x}_t, \tilde{y}_t)$, where $\tilde{y}_t$ is the truncated reward $y_t \mathbb{I}_{\|\boldsymbol{x}_t\|_{\mathbf{V}_t^{-1}}|y_t| \le \Gamma}$. Replacing the $(\boldsymbol{x}_t, y_t)$ of general upper bound (15) by $(\boldsymbol{x}_t, \tilde{y}_t)$, we get that

$$
\|\hat{\boldsymbol{\theta}}_{t+1} - \boldsymbol{\theta}_*\|_{\mathbf{V}_{t+1}}^2
$$

$$
\le \|\hat{\boldsymbol{\theta}}_1 - \boldsymbol{\theta}_*\|_{\mathbf{V}_1}^2 - \frac{\kappa}{2} \sum_{\tau=1}^{t} \left(\boldsymbol{x}_\tau^\top \hat{\boldsymbol{\theta}}_\tau - \boldsymbol{x}_\tau^\top \boldsymbol{\theta}_*\right)^2 + \sum_{\tau=1}^{t} \left(\mu(\boldsymbol{x}_\tau^\top \boldsymbol{\theta}_*) - \mu(\boldsymbol{x}_\tau^\top \hat{\boldsymbol{\theta}}_\tau)\right)^2 \|\boldsymbol{x}_\tau\|_{\mathbf{V}_\tau^{-1}}^2 \quad (16)
$$

$$
+ \sum_{\tau=1}^{t} 2\left(\tilde{y}_\tau - \mu(\boldsymbol{x}_\tau^\top \boldsymbol{\theta}_*)\right) \boldsymbol{x}_\tau^\top \left(\hat{\boldsymbol{\theta}}_\tau - \boldsymbol{\theta}_*\right) + \sum_{\tau=1}^{t} \left(\tilde{y}_\tau - \mu(\boldsymbol{x}_\tau^\top \boldsymbol{\theta}_*)\right)^2 \|\boldsymbol{x}_\tau\|_{\mathbf{V}_\tau^{-1}}^2.
$$

Assumption 2 shows that the upper bound of $\mu(\cdot)$ is $U$. Thus, the inequality (16) can be simplified as

$$
\|\hat{\boldsymbol{\theta}}_{t+1} - \boldsymbol{\theta}_*\|_{\mathbf{V}_{t+1}}^2 \le \|\hat{\boldsymbol{\theta}}_1 - \boldsymbol{\theta}_*\|_{\mathbf{V}_1}^2 + 6U^2 \sum_{\tau=1}^{t} \|\boldsymbol{x}_\tau\|_{\mathbf{V}_\tau^{-1}}^2
$$

$$
+ 2 \sum_{\tau=1}^{t} \left(\tilde{y}_\tau - \mu(\boldsymbol{x}_\tau^\top \boldsymbol{\theta}_*)\right) \boldsymbol{x}_\tau^\top \left(\hat{\boldsymbol{\theta}}_\tau - \boldsymbol{\theta}_*\right)
$$

$$
+ 2 \sum_{\tau=1}^{t} \|\boldsymbol{x}_\tau\|_{\mathbf{V}_\tau^{-1}}^2 y_\tau^2 \mathbb{I}_{\|\boldsymbol{x}_\tau\|_{\mathbf{V}_\tau^{-1}}|y_\tau| \le \Gamma}.
$$

We define $\beta_\tau = \|\boldsymbol{x}_\tau\|_{\mathbf{V}_\tau^{-1}}$. Since $\mathbf{V}_1 = \lambda \mathbf{I}_d$ and $\hat{\boldsymbol{\theta}}_1 = \mathbf{0}$, we can deduce that

$$
\|\hat{\boldsymbol{\theta}}_{t+1} - \boldsymbol{\theta}_*\|_{\mathbf{V}_{t+1}}^2 \le \lambda S^2 + 6U^2 \sum_{\tau=1}^{t} \beta_\tau^2 + 2 \underbrace{\sum_{\tau=1}^{t} \beta_\tau^2 y_\tau^2 \mathbb{I}_{|\beta_\tau y_\tau| \le \Gamma}}_{A}
$$

$$
+ 2 \sum_{\tau=1}^{t} \underbrace{\left(y_\tau \mathbb{I}_{|\beta_\tau y_\tau| \le \Gamma} - \mu(\boldsymbol{x}_\tau^\top \boldsymbol{\theta}_*)\right) \boldsymbol{x}_\tau^\top \left(\hat{\boldsymbol{\theta}}_\tau - \boldsymbol{\theta}_*\right)}_{B_\tau}. \quad (17)
$$

Then, we will employ analytic techniques of truncated strategy to bound the terms $A$ and $\sum_{\tau=1}^{t} B_\tau$.

**Lemma 4** *Suppose that* $\mathrm{E}\left[|y_\tau|^{1+\epsilon}|\mathcal{G}_{\tau-1}\right] \le u$ *for* $\tau = 1, 2, \ldots, t$. *Then, we have that*

$$
A \le 2\Gamma^2 \ln(2/\delta) + \frac{3}{2}\Gamma^{1-\epsilon} \sum_{\tau=1}^{t} \beta_\tau^{1+\epsilon} u
$$

*holds with probability at least* $1 - \delta$.

**Proof.** According to the triangle inequality, $A$ can be relaxed as

$$
A \le \left|\sum_{\tau=1}^{t} \beta_\tau^2 y_\tau^2 \mathbb{I}_{|\beta_\tau y_\tau| \le \Gamma} - \mathrm{E}\left[\beta_\tau^2 y_\tau^2 \mathbb{I}_{|\beta_\tau y_\tau| \le \Gamma}|\mathcal{G}_{\tau-1}\right]\right| + \sum_{\tau=1}^{t} \mathrm{E}\left[\beta_\tau^2 y_\tau^2 \mathbb{I}_{|\beta_\tau y_\tau| \le \Gamma}|\mathcal{G}_{\tau-1}\right]. \quad (18)
$$

In light of Bernstein's inequality [Seldin *et al.*, 2012, Lemma 11], we have that

$$
\left|\sum_{\tau=1}^{t} \beta_\tau^2 y_\tau^2 \mathbb{I}_{|\beta_\tau y_\tau| \le \Gamma} - \mathrm{E}\left[\beta_\tau^2 y_\tau^2 \mathbb{I}_{|\beta_\tau y_\tau| \le \Gamma}|\mathcal{G}_{\tau-1}\right]\right|
$$

$$
\le 2\Gamma^2 \ln(2/\delta) + \frac{1}{2\Gamma^2} \sum_{\tau=1}^{t} \mathrm{Var}\left[\beta_\tau^2 y_\tau^2 \mathbb{I}_{|\beta_\tau y_\tau| \le \Gamma}|\mathcal{G}_{\tau-1}\right] \quad (19)
$$

$$
\le 2\Gamma^2 \ln(2/\delta) + \frac{1}{2\Gamma^2} \sum_{\tau=1}^{t} \beta_\tau^{1+\epsilon} u \Gamma^{3-\epsilon}
$$

holds with probability at least $1 - \delta$, and the second inequality of above equation holds because the $(1 + \epsilon)$-th moment of rewards is bounded by $u$.

We can bound the second term in the right side of (18) as

$$\sum_{\tau=1}^{t} \mathrm{E}\left[\beta_\tau^2 y_\tau^2 \mathbb{I}_{|\beta_\tau y_\tau| \leq \Gamma} | \mathcal{G}_{\tau-1}\right] \leq \Gamma^{1-\epsilon} \sum_{\tau=1}^{t} \beta_\tau^{1+\epsilon} u. \tag{20}$$

Combining the inequalities (18), (19) and (20) finishes the proof of Lemma 4. $\qquad\square$

We will now proceed to bound the term $\sum_{\tau=1}^{t} B_\tau$.

**Lemma 5** *Suppose that* $\mathrm{E}\left[|y_\tau|^{1+\epsilon} | \mathcal{G}_{\tau-1}\right] \leq u$ *for* $\tau = 1, 2, \ldots, t$. *Then, we have that*

$$\sum_{\tau=1}^{t} B_\tau \leq 2\Gamma\gamma^{\frac{1}{2}} \ln(2/\delta) + \frac{3\gamma^{\frac{1}{2}}}{2\Gamma^\epsilon} \sum_{\tau=1}^{t} \beta_\tau^{1+\epsilon} u$$

*holds with probability at least* $1 - T\delta$.

**Proof.** First, we give the fact that

$$\left| \boldsymbol{x}_\tau^\top (\hat{\boldsymbol{\theta}}_\tau - \boldsymbol{\theta}_*) y_\tau \mathbb{I}_{|\beta_\tau y_\tau| \leq \Gamma} \right| \leq \|\hat{\boldsymbol{\theta}}_\tau - \boldsymbol{\theta}_*\|_{\mathbf{V}_\tau} \|\boldsymbol{x}_\tau\|_{\mathbf{V}_\tau^{-1}} |y_\tau| \mathbb{I}_{|\beta_\tau y_\tau| \leq \Gamma}$$
$$\leq \|\hat{\boldsymbol{\theta}}_\tau - \boldsymbol{\theta}_*\|_{\mathbf{V}_\tau} \Gamma.$$

Then, through the full probability formula [Mendenhall *et al.*, 2012], we have that

$$\mathbb{P}\left\{ \sum_{\tau=1}^{t} B_\tau > \chi \right\} \leq \mathbb{P}\left\{ \exists \tau, \|\hat{\boldsymbol{\theta}}_\tau - \boldsymbol{\theta}_*\|_{\mathbf{V}_\tau}^2 \geq \gamma \right\} + \mathbb{P}\left\{ \sum_{\tau=1}^{t} B_\tau \mathbb{I}_{\|\hat{\boldsymbol{\theta}}_\tau - \boldsymbol{\theta}_*\|_{\mathbf{V}_\tau}^2 \leq \gamma} > \chi \right\}$$
$$\leq (T-1)\delta + \mathbb{P}\left\{ \sum_{\tau=1}^{t} B_\tau \mathbb{I}_{\|\hat{\boldsymbol{\theta}}_\tau - \boldsymbol{\theta}_*\|_{\mathbf{V}_\tau}^2 \leq \gamma} > \chi \right\}. \tag{21}$$

The second inequality of above equation holds because $\|\hat{\boldsymbol{\theta}}_\tau - \boldsymbol{\theta}_*\|_{\mathbf{V}_\tau}^2 \geq \gamma$ with probability at most $\delta$.

In the following, we analyze the term $\sum_{\tau=1}^{t} B_\tau \mathbb{I}_{\|\hat{\boldsymbol{\theta}}_\tau - \boldsymbol{\theta}_*\|_{\mathbf{V}_\tau}^2 \leq \gamma}$ to determine the appropriate $\chi$ for bounding the right side of (21). A simple application of the triangle inequality shows that

$$\sum_{\tau=1}^{t} B_\tau \mathbb{I}_{\|\hat{\boldsymbol{\theta}}_\tau - \boldsymbol{\theta}_*\|_{\mathbf{V}_\tau}^2 \leq \gamma} \leq \left| \sum_{\tau=1}^{t} (\tilde{y}_\tau - \mathrm{E}[\tilde{y}_\tau | \mathcal{G}_{\tau-1}]) \boldsymbol{x}_\tau^\top (\hat{\boldsymbol{\theta}}_\tau - \boldsymbol{\theta}_*) \mathbb{I}_{\|\hat{\boldsymbol{\theta}}_\tau - \boldsymbol{\theta}_*\|_{\mathbf{V}_\tau}^2 \leq \gamma} \right|$$
$$+ \left| \sum_{\tau=1}^{t} \mathrm{E}[y_\tau \mathbb{I}_{|\beta_\tau y_\tau| \geq \Gamma} | \mathcal{G}_{\tau-1}] \boldsymbol{x}_\tau^\top (\hat{\boldsymbol{\theta}}_\tau - \boldsymbol{\theta}_*) \mathbb{I}_{\|\hat{\boldsymbol{\theta}}_\tau - \boldsymbol{\theta}_*\|_{\mathbf{V}_\tau}^2 \leq \gamma} \right| \tag{22}$$

By utilizing Bernstein's inequality [Seldin *et al.*, 2012, Lemma 11], we can demonstrate that,

$$\left| \sum_{\tau=1}^{t} (\tilde{y}_\tau - \mathrm{E}[\tilde{y}_\tau | \mathcal{G}_{\tau-1}]) \boldsymbol{x}_\tau^\top (\hat{\boldsymbol{\theta}}_\tau - \boldsymbol{\theta}_*) \mathbb{I}_{\|\hat{\boldsymbol{\theta}}_\tau - \boldsymbol{\theta}_*\|_{\mathbf{V}_\tau}^2 \leq \gamma} \right|$$
$$\leq \frac{1}{2\Gamma\gamma^{\frac{1}{2}}} \sum_{\tau=1}^{t} \mathrm{Var}[\boldsymbol{x}_\tau^\top (\hat{\boldsymbol{\theta}}_\tau - \boldsymbol{\theta}_*) \mathbb{I}_{\|\hat{\boldsymbol{\theta}}_\tau - \boldsymbol{\theta}_*\|_{\mathbf{V}_\tau}^2 \leq \gamma} \tilde{y}_\tau | \mathcal{G}_{\tau-1}] + 2\Gamma\gamma^{\frac{1}{2}} \ln(2/\delta).$$

holds with probability at least $1 - \delta$. Additionally, apply the Cauchy-Schwarz inequality and the scalar property of variance, we can establish that

$$\mathrm{Var}[\boldsymbol{x}_\tau^\top (\hat{\boldsymbol{\theta}}_\tau - \boldsymbol{\theta}_*) \mathbb{I}_{\|\hat{\boldsymbol{\theta}}_\tau - \boldsymbol{\theta}_*\|_{\mathbf{V}_\tau}^2 \leq \gamma} \tilde{y}_\tau | \mathcal{G}_{\tau-1}] \leq \gamma \cdot \mathrm{Var}[\beta_\tau \tilde{y}_\tau | \mathcal{G}_{\tau-1}].$$

Thus, we have that

$$\left| \sum_{\tau=1}^{t} (\tilde{y}_\tau - \mathrm{E}[\tilde{y}_\tau | \mathcal{G}_{\tau-1}]) \boldsymbol{x}_\tau^\top (\hat{\boldsymbol{\theta}}_\tau - \boldsymbol{\theta}_*) \mathbb{I}_{\|\hat{\boldsymbol{\theta}}_\tau - \boldsymbol{\theta}_*\|_{\mathbf{V}_\tau}^2 \leq \gamma} \right|$$

$$\leq \frac{\gamma}{2\Gamma\gamma^{\frac{1}{2}}} \sum_{\tau=1}^{t} \mathrm{Var}[\beta_\tau y_\tau \mathbb{I}_{|\beta_\tau y_\tau| \leq \Gamma} | \mathcal{G}_{\tau-1}] + 2\Gamma\gamma^{\frac{1}{2}} \ln(2/\delta) \tag{23}$$

$$\leq \frac{\gamma^{\frac{1}{2}}}{2\Gamma^\epsilon} \sum_{\tau=1}^{t} \beta_\tau^{1+\epsilon} u + 2\Gamma\gamma^{\frac{1}{2}} \ln(2/\delta).$$

The second term on the right side of inequality (22) can be bounded as

$$\left| \sum_{\tau=1}^{t} \mathrm{E}[y_\tau \mathbb{I}_{|\beta_\tau y_\tau| \geq \Gamma} | \mathcal{G}_{\tau-1}] \boldsymbol{x}_\tau^\top (\hat{\boldsymbol{\theta}}_\tau - \boldsymbol{\theta}_*) \mathbb{I}_{\|\hat{\boldsymbol{\theta}}_\tau - \boldsymbol{\theta}_*\|_{\mathbf{V}_\tau}^2 \leq \gamma} \right|$$

$$\leq \gamma^{\frac{1}{2}} \sum_{\tau=1}^{t} \mathrm{E}[|\beta_\tau y_\tau| \mathbb{I}_{|\beta_\tau y_\tau| \geq \Gamma} | \mathcal{G}_{\tau-1}] \leq \frac{\gamma^{\frac{1}{2}}}{\Gamma^\epsilon} \sum_{\tau=1}^{t} \beta_\tau^{1+\epsilon} u. \tag{24}$$

Taking (23), (24) into (22), we have the inequality

$$\sum_{\tau=1}^{t} B_\tau \mathbb{I}_{\|\hat{\boldsymbol{\theta}}_\tau - \boldsymbol{\theta}_*\|_{\mathbf{V}_\tau}^2 \leq \gamma} \leq \frac{3\gamma^{\frac{1}{2}}}{2\Gamma^\epsilon} \sum_{\tau=1}^{t} \beta_\tau^{1+\epsilon} u + 2\Gamma\gamma^{\frac{1}{2}} \ln(2/\delta).$$

holds with probability at least $1 - \delta$. Let $\chi$ of inequality (21) be $2\Gamma\gamma^{\frac{1}{2}} \ln(2/\delta) + \frac{3\gamma^{\frac{1}{2}}}{2\Gamma^\epsilon} \sum_{\tau=1}^{t} \beta_\tau^{1+\epsilon} u$, we have

$$\mathbb{P} \left\{ \sum_{\tau=1}^{t} B_\tau > \chi \right\} \leq T\delta.$$

The proof of Lemma 5 is finished. $\qquad\square$

We have bounded the terms $A$ and $\sum_{\tau=1}^{t} B_\tau$ using Lemma 4 and Lemma 5, respectively. By incorporating these two lemmas into equation (17) and substituting $\delta$ with $\delta/2T$, we can derive that

$$\|\hat{\boldsymbol{\theta}}_{t+1} - \boldsymbol{\theta}_*\|_{\mathbf{V}_{t+1}}^2 \leq 6U^2 \sum_{\tau=1}^{t} \beta_\tau^2 + 4\Gamma^2 \ln(4T/\delta) + 4\Gamma\gamma^{\frac{1}{2}} \ln(4T/\delta)$$

$$+ \lambda S^2 + 3\Gamma^{1-\epsilon} \sum_{\tau=1}^{t} \beta_\tau^{1+\epsilon} v + 3\gamma^{\frac{1}{2}} \Gamma^{-\epsilon} \sum_{\tau=1}^{t} \beta_\tau^{1+\epsilon} u. \tag{25}$$

holds with probability at least $1 - \delta$. The Hölder inequality tells that

$$\sum_{\tau=1}^{t} \beta_\tau^{1+\epsilon} \leq t^{\frac{1-\epsilon}{2}} \left( \sum_{\tau=1}^{t} \beta_\tau^2 \right)^{\frac{1+\epsilon}{2}}. \tag{26}$$

Then, according to Lemma 11 of Abbasi-yadkori *et al.* [2011], we have that

$$\sum_{\tau=1}^{T} \beta_\tau^2 = \sum_{\tau=1}^{T} \|\boldsymbol{x}_\tau\|_{\mathbf{V}_\tau^{-1}}^2 \leq \frac{4}{\kappa} \ln\left( \frac{\det(\mathbf{V}_{T+1})}{\det(\mathbf{V}_1)} \right) \leq \frac{4d}{\kappa} \ln\left( 1 + \frac{\kappa T}{2\lambda d} \right).$$

Thus, (26) can be relaxed as

$$\sum_{\tau=1}^{t} \beta_\tau^{1+\epsilon} \leq T^{\frac{1-\epsilon}{2}} \left( \frac{4d}{\kappa} \ln\left( 1 + \frac{\kappa T}{2\lambda d} \right) \right)^{\frac{1+\epsilon}{2}}. \tag{27}$$

By taking (27) into (25) and let

$$\Gamma = 2(u \ln(4T/\delta))^{\frac{1}{1+\epsilon}} \left( d\kappa \ln\left( 1 + \frac{\kappa T}{2\lambda d} \right) \right)^{\frac{1}{2}} T^{\frac{1-\epsilon}{2(1+\epsilon)}},$$

we have that

$$\|\hat{\boldsymbol{\theta}}_{t+1} - \boldsymbol{\theta}_*\|^2_{\mathbf{V}_{t+1}} \leq \lambda S^2 + \frac{24U^2 d}{\kappa} \ln\left(1 + \frac{\kappa T}{2\lambda d}\right)$$

$$+ 7u^{\frac{2}{1+\epsilon}} \ln(4T/\delta)^{\frac{\epsilon-1}{1+\epsilon}} T^{\frac{1-\epsilon}{1+\epsilon}} \frac{4d}{\kappa} \ln\left(1 + \frac{\kappa T}{2\lambda d}\right) \tag{28}$$

$$+ 7u^{\frac{1}{1+\epsilon}} \ln(4T/\delta)^{\frac{\epsilon}{1+\epsilon}} T^{\frac{1-\epsilon}{2(1+\epsilon)}} \gamma^{\frac{1}{2}} \left(\frac{4d}{\kappa} \ln\left(1 + \frac{\kappa T}{2\lambda d}\right)\right)^{\frac{1}{2}}$$

holds with probability at least $1 - \delta$.

In order to determine $\gamma$ satisfying $\|\hat{\boldsymbol{\theta}}_{t+1} - \boldsymbol{\theta}_*\|^2_{\mathbf{V}_{t+1}} \leq \gamma$, a quadratic inequality with respect to $\gamma$ need to be solved, such that the right side of inequality (28) is smaller than $\gamma$. This leads to the conclusion that

$$\gamma = 112v^{\frac{2}{1+\epsilon}} \ln(4T/\delta)^{\frac{2\epsilon}{1+\epsilon}} T^{\frac{1-\epsilon}{1+\epsilon}} \frac{4d}{\kappa} \ln\left(1 + \frac{\kappa T}{2\lambda d}\right) + 2\lambda S^2 + \frac{48U^2 d}{\kappa} \ln\left(1 + \frac{\kappa T}{2\lambda d}\right).$$

By taking the union bound over all $t$, we have that with probability at least $1 - \delta$, for any $t > 0$, the inequality

$$\|\hat{\boldsymbol{\theta}}_{t+1} - \boldsymbol{\theta}_*\|^2_{\mathbf{V}_{t+1}} \leq 224v^{\frac{2}{1+\epsilon}} \ln(4T/\delta)^{\frac{2\epsilon}{1+\epsilon}} T^{\frac{1-\epsilon}{1+\epsilon}} \frac{4d}{\kappa} \ln\left(1 + \frac{\kappa T}{2\lambda d}\right)$$

$$+ 2\lambda S^2 + \frac{48U^2 d}{\kappa} \ln\left(1 + \frac{\kappa T}{2\lambda d}\right)$$

holds, which concludes the proof of Theorem 1. $\qquad\square$

## B  Proof of Theorem 2

To begin with, we bound the instantaneous regret by the following lemma.

**Lemma 6** *If $\boldsymbol{\theta}_* \in \mathcal{C}_t$ for all t, then*

$$\mu(\tilde{\boldsymbol{x}}_t^\top \boldsymbol{\theta}_*) - \mu(\boldsymbol{x}_t^\top \boldsymbol{\theta}_*) \leq 2L\sqrt{\gamma_t}\|\boldsymbol{x}_t\|_{\mathbf{V}_t^{-1}}$$

*where $\tilde{\boldsymbol{x}}_t = \mathrm{argmax}_{\boldsymbol{x} \in \mathcal{D}_t} \mu(\boldsymbol{x}^\top \boldsymbol{\theta}_*)$.*

**Proof.** Considering that the link function $\mu(\cdot)$ is $L$-Lipschitz and monotonically increasing, we have

$$\begin{aligned}
\mu(\tilde{\boldsymbol{x}}_t^\top \boldsymbol{\theta}_*) - \mu(\boldsymbol{x}_t^\top \boldsymbol{\theta}_*) &\leq \max\{0, L(\tilde{\boldsymbol{x}}_t^\top \boldsymbol{\theta}_* - \boldsymbol{x}_t^\top \boldsymbol{\theta}_*)\} \\
&\leq \max\{0, L(\boldsymbol{x}_t^\top \tilde{\boldsymbol{\theta}}_t - \boldsymbol{x}_t^\top \boldsymbol{\theta}_*)\} \\
&= \max\{0, L\boldsymbol{x}_t^\top(\tilde{\boldsymbol{\theta}}_t - \hat{\boldsymbol{\theta}}_t) + L\boldsymbol{x}_t^\top(\hat{\boldsymbol{\theta}}_t - \boldsymbol{\theta}_*)\} \\
&\leq L\left(\|\tilde{\boldsymbol{\theta}}_t - \hat{\boldsymbol{\theta}}_t\|_{\mathbf{V}_t} + \|\hat{\boldsymbol{\theta}}_t - \boldsymbol{\theta}_*\|_{\mathbf{V}_t}\right)\|\boldsymbol{x}_t\|_{\mathbf{V}_t^{-1}} \\
&\leq 2L\sqrt{\gamma_t}\|\boldsymbol{x}_t\|_{\mathbf{V}_t^{-1}}
\end{aligned}$$

where the second inequality holds due to the fact that $(\boldsymbol{x}_t, \tilde{\boldsymbol{\theta}}_t) = \mathrm{argmax}_{\boldsymbol{x} \in \mathcal{D}_t, \boldsymbol{\theta} \in \mathcal{C}_t}\langle \boldsymbol{x}, \boldsymbol{\theta}\rangle$. $\qquad\square$

Then, we get the regret of CRTM through the cumulative summation from 1 to $T$.

**Lemma 7** *If $\boldsymbol{\theta}_* \in \mathcal{C}_t$ for all t, then the regret of CRTM can be bounded as*

$$R(T) \leq 2L\left(\frac{4d}{\kappa} \ln\left(1 + \frac{\kappa T}{2\lambda d}\right) \sum_{t=1}^T \gamma_t\right)^{1/2}.$$

**Proof.** Through the Lemma 6, we have that

$$\begin{aligned}
R(T) = \sum_{t=1}^T \mu(\tilde{\boldsymbol{x}}_t^\top \boldsymbol{\theta}_*) - \mu(\boldsymbol{x}_t^\top \boldsymbol{\theta}_*) &\leq 2L\sum_{t=1}^T \sqrt{\gamma_t}\|\boldsymbol{x}_t\|_{\mathbf{V}_t^{-1}} \\
&\leq 2L\left(\sum_{t=1}^T \gamma_t\right)^{1/2}\left(\sum_{t=1}^T \|\boldsymbol{x}_t\|^2_{\mathbf{V}_t^{-1}}\right)^{1/2}.
\end{aligned} \tag{29}$$

According to the Lemma 11 of Abbasi-yadkori *et al.* [2011], we get that

$$\sum_{t=1}^{T} \|\boldsymbol{x}_t\|_{\mathbf{V}_t^{-1}}^2 \leq \frac{4}{\kappa} \ln \left( \frac{\det(\mathbf{V}_{T+1})}{\det(\mathbf{V}_1)} \right) \leq \frac{4d}{\kappa} \ln \left( 1 + \frac{\kappa T}{2\lambda d} \right). \tag{30}$$

Combining (29) and (30) finishes the proof. $\qquad\qquad\qquad\qquad\qquad\qquad\qquad\qquad\square$

By substituting $\gamma$ of Theorem 1 into Lemma 7 such that $\gamma_t = \gamma$ for $t = 1, 2, \ldots, T$, the regret bound of CRTM is explicitly given as

$$
\begin{aligned}
R(T) \leq\;& 128 L \kappa^{-1} v^{\frac{1}{1+\epsilon}} d \ln(4T/\delta)^{\frac{\epsilon}{1+\epsilon}} \ln \left( 1 + \frac{\kappa T}{2\lambda d} \right) T^{\frac{1}{1+\epsilon}} \\
&+ 24 L U \kappa^{-1} d \ln \left( 1 + \frac{\kappa T}{2\lambda d} \right) T^{\frac{1}{2}} \\
&+ 8 L S (\lambda d)^{\frac{1}{2}} \kappa^{-\frac{1}{2}} \left( \ln \left( 1 + \frac{\kappa T}{2\lambda d} \right) \right)^{\frac{1}{2}} T^{\frac{1}{2}} \\
=\;& O \left( d(\log T)^{\frac{1+2\epsilon}{1+\epsilon}} T^{\frac{1}{1+\epsilon}} \right).
\end{aligned}
$$

The proof of Theorem 2 is finished.

## C   Proof of Theorem 3

Notice that CRMM updates the estimator with action-reward pair $(\boldsymbol{x}_t, \bar{y}_t)$, where $\bar{y}_t$ is the median of $\{y_t^1, y_t^2, \ldots, y_t^r\}$. Replace $(\boldsymbol{x}_t, y_t)$ of general upper bound (15) by $(\boldsymbol{x}_t, \bar{y}_t)$, we get that

$$
\begin{aligned}
\|\hat{\boldsymbol{\theta}}_{t+1} - \boldsymbol{\theta}_*\|_{\mathbf{V}_{t+1}}^2 \leq\;& \|\hat{\boldsymbol{\theta}}_1 - \boldsymbol{\theta}_*\|_{\mathbf{V}_1}^2 - \frac{\kappa}{2} \sum_{\tau=1}^{t} \alpha_\tau^2 + \sum_{\tau=1}^{t} \left( \mu(\boldsymbol{x}_\tau^\top \boldsymbol{\theta}_*) - \mu(\boldsymbol{x}_\tau^\top \hat{\boldsymbol{\theta}}_\tau) \right)^2 \|\boldsymbol{x}_\tau\|_{\mathbf{V}_\tau^{-1}}^2 \\
&+ \sum_{\tau=1}^{t} 2\boldsymbol{x}_\tau^\top (\hat{\boldsymbol{\theta}}_\tau - \boldsymbol{\theta}_*) \left( \bar{y}_\tau - \mu(\boldsymbol{x}_\tau^\top \boldsymbol{\theta}_*) \right) + \sum_{\tau=1}^{t} \|\boldsymbol{x}_\tau\|_{\mathbf{V}_\tau^{-1}}^2 \left( \bar{y}_\tau - \mu(\boldsymbol{x}_\tau^\top \boldsymbol{\theta}_*) \right)^2.
\end{aligned}
$$

Let $\alpha_\tau = \boldsymbol{x}_\tau^\top \left( \hat{\boldsymbol{\theta}}_\tau - \boldsymbol{\theta}_* \right), \beta_\tau = \|\boldsymbol{x}_\tau\|_{\mathbf{V}_\tau^{-1}}$ and $X_\tau = \bar{y}_\tau - \mu(\boldsymbol{x}_\tau^\top \boldsymbol{\theta}_*)$. The above equation can be simplified as

$$
\begin{aligned}
\|\hat{\boldsymbol{\theta}}_{t+1} - \boldsymbol{\theta}_*\|_{\mathbf{V}_{t+1}}^2 \leq\;& \|\hat{\boldsymbol{\theta}}_1 - \boldsymbol{\theta}_*\|_{\mathbf{V}_1}^2 - \frac{\kappa}{2} \sum_{\tau=1}^{t} \alpha_\tau^2 + \sum_{\tau=1}^{t} \left( \mu(\boldsymbol{x}_\tau^\top \boldsymbol{\theta}_*) - \mu(\boldsymbol{x}_\tau^\top \hat{\boldsymbol{\theta}}_\tau) \right)^2 \|\boldsymbol{x}_\tau\|_{\mathbf{V}_\tau^{-1}}^2 \\
&+ \sum_{\tau=1}^{t} 2\alpha_\tau X_\tau + \sum_{\tau=1}^{t} \beta_\tau^2 X_\tau^2.
\end{aligned} \tag{31}
$$

We need to bound the terms $\sum_{\tau=1}^{t} \alpha_\tau X_\tau$ and $\sum_{\tau=1}^{t} \beta_\tau^2 X_\tau^2$ to conduct a narrow confidence region. Considering that the latent idea of CRMM is mean of medians, we provide the following lemma to display the $(1 + \epsilon)$-th moment for the median term.

**Lemma 8** *Suppose $X^1, \ldots, X^r$ are independently drawn from the distribution $\chi$, and $\mathrm{E}[X^i] = 0$, $\mathrm{E}[|X^i|^{1+\epsilon}] \leq v$ for $i = 1, 2, \ldots, r$. If $\widehat{X}$ is the median of $\{X^i\}_{i=1}^r$, then $\widehat{X}$ satisfies $\mathrm{E}[|\widehat{X}|^{1+\epsilon}] \leq rv$.*

**Proof.** Let the p.d.f and c.d.f of $\chi$ be denoted as $p(x)$ and $F(x)$, respectively. Then, the c.d.f of $\widehat{X}$ can be calculated as

$$\mathbb{P}\{\widehat{X} \leq x\} = \sum_{k=\lceil r/2 \rceil}^{r} \binom{r}{k} F(x)^k (1 - F(x))^{r-k}.$$

Taking the derivative of the above equation, the p.d.f of $\widehat{X}$ can be obtained as

$$f(x) = r\binom{r-1}{\lceil r/2 \rceil - 1} F(x)^{\lceil r/2 \rceil - 1}(1 - F(x))^{r - \lceil r/2 \rceil}p(x).$$

According to the fact $\binom{r-1}{\lceil r/2 \rceil - 1} F(x)^{\lceil r/2 \rceil - 1}(1 - F(x))^{r - \lceil r/2 \rceil} \leq 1$, we can easily get that

$$f(x) \leq rp(x).$$

Thus, the $(1 + \epsilon)$-th moment of $\widehat{X}$ satisfies

$$\mathrm{E}[|\widehat{X}|^{1+\epsilon}] = \int |x|^{1+\epsilon} f(x)\mathrm{d} \leq r \int |x|^{1+\epsilon}p(x)\mathrm{d} \leq rv.$$

The proof of Lemma 8 is finished. $\qquad\square$

Another tool used to bound $\sum_{\tau=1}^{t} \alpha_\tau X_\tau$ is displayed as follows, whose proof is provided in Section E.

**Lemma 9** *Suppose that $X_1, \ldots, X_n$ are random variables satisfying $\mathrm{E}[X_i|\mathcal{F}_{i-1}] = 0$, and $\mathrm{E}[|X_i|^{1+\epsilon}|\mathcal{F}_{i-1}] \leq v_1$, where $\mathcal{F}_{i-1} \triangleq \{X_1, \ldots, X_{i-1}\}$ is a $\sigma$-filtration and $\mathcal{F}_0 = \emptyset$. For the fixed parameters $\alpha_1, \alpha_2, \ldots, \alpha_n \in \mathbb{R}$ and $C > 0$, with probability at least $1 - \delta$, we have that*

$$\left| \sum_{i=1}^{n} \alpha_i X_i \mathbb{I}_{|\alpha_i X_i| \leq C\|\boldsymbol{\alpha}\|_{1+\epsilon}} \right| \leq \xi\|\boldsymbol{\alpha}\|_{1+\epsilon}$$

*where*

$$\boldsymbol{\alpha} = [\alpha_1, \alpha_2, \ldots, \alpha_n], \xi = 2C\ln(2/\delta) + 2C^{-\epsilon}v_1.$$

Equipped with Lemma 8 and Lemma 9, we are ready to bound the term $\sum_{\tau=1}^{t} \alpha_\tau X_\tau$.

**Lemma 10** *Let $r = \lceil 16\ln\frac{4T}{\delta} \rceil$, for any $t > 0$, with probability at least $1 - \delta/T$,*

$$\sum_{\tau=1}^{t} \alpha_\tau X_\tau \leq \rho\|\boldsymbol{\alpha}\|_{1+\epsilon}$$

*where*

$$\boldsymbol{\alpha} = [\alpha_1, \alpha_2, \ldots, \alpha_t], C = (4v)^{\frac{1}{1+\epsilon}}, \rho = 2C\ln(4T/\delta) + 2C^{-\epsilon}rv.$$

**Proof.** Through the full probability formula [Mendenhall *et al.*, 2012], we have that

$$\Pr\left\{ \left| \sum_{i=1}^{t} \alpha_\tau X_\tau \right| > \rho\|\boldsymbol{\alpha}\|_{1+\epsilon} \right\} \leq \Pr\left\{ \left| \sum_{\tau=1}^{t} \alpha_\tau X_\tau \mathbb{I}_{|\alpha_\tau X_\tau| \leq C\|\boldsymbol{\alpha}\|_{1+\epsilon}} \right| > \rho\|\boldsymbol{\alpha}\|_{1+\epsilon} \right\} \qquad (32)$$
$$+ \sum_{\tau=1}^{t} \Pr\left\{ |\alpha_\tau X_\tau| > C\|\boldsymbol{\alpha}\|_{1+\epsilon} \right\}$$

We first analyze the second term in the right side of (32). Recall that CRMM observes $r$ rewards $\{y_\tau^1, \ldots, y_\tau^r\}$ at round $\tau$, and for the sake of representation, we denote the difference between $y_\tau^i$ and $\mu(\boldsymbol{x}_\tau^\top \boldsymbol{\theta}_*)$ as $X_\tau^i$, such that $X_\tau^i = y_\tau^i - \mu(\boldsymbol{x}_\tau^\top \boldsymbol{\theta}_*)$. Through Markov's inequality and the heavy-tailed condition $\mathrm{E}[|X_\tau^i|^{1+\epsilon}] \leq v$, we have that

$$\Pr\left\{ |\alpha_\tau X_\tau^i| > C\|\boldsymbol{\alpha}\|_{1+\epsilon} \right\} \leq \frac{|\alpha_\tau|^{1+\epsilon}v}{C^{1+\epsilon}\|\boldsymbol{\alpha}\|_{1+\epsilon}^{1+\epsilon}}.$$

Let $C = (4v)^{\frac{1}{1+\epsilon}}$, then we have

$$\Pr\left\{ |\alpha_\tau X_\tau^i| > C\|\boldsymbol{\alpha}\|_{1+\epsilon} \right\} \leq \frac{1}{4}.$$

Define the random variables

$$B_i = \mathbb{I}_{\alpha_\tau X_\tau^i > C\|\boldsymbol{\alpha}\|_{1+\epsilon}},$$

thus $p_i = \Pr\{B_i = 1\} \le \frac{1}{4}$. According to the Azuma-Hoeffing's inequality [Azuma, 1967], we get

$$\Pr\left\{\sum_{i=1}^{r} B_j \ge \frac{r}{2}\right\} \le \Pr\left\{\sum_{i=1}^{r} B_i - p_i \ge \frac{r}{4}\right\}$$

$$\le e^{-r/8} \le \frac{\delta}{4T^2}$$

for $r = \left\lceil 16 \ln \frac{4T}{\delta} \right\rceil$. The inequality $\sum_{i=1}^{r} B_i \ge \frac{r}{2}$ means more than half of the terms $\{B_i\}_{i=1}^{r}$ is true. Thus, the median term $\alpha_\tau X_\tau$ satisfies

$$\alpha_\tau X_\tau > C\|\boldsymbol{\alpha}\|_{1+\epsilon}$$

with probability at most $\frac{\delta}{4T^2}$. A similar argument shows that

$$\alpha_\tau X_\tau < -C\|\boldsymbol{\alpha}\|_{1+\epsilon}$$

holds with probability at most $\frac{\delta}{4T^2}$. Therefore, we have

$$\Pr\left\{|\alpha_\tau X_\tau| > C\|\boldsymbol{\alpha}\|_{1+\epsilon}\right\} \le \frac{\delta}{2T^2}.$$

By taking it into (32), we have that

$$\Pr\left\{\left|\sum_{\tau=1}^{t} \alpha_\tau X_\tau\right| > \rho\|\boldsymbol{\alpha}\|_{1+\epsilon}\right\} \le \frac{\delta}{2T} + \Pr\left\{\left|\sum_{\tau=1}^{t} \alpha_\tau X_\tau \mathbb{I}_{|\alpha_\tau X_\tau| \le C\|\boldsymbol{\alpha}\|_{1+\epsilon}}\right| > \rho\|\boldsymbol{\alpha}\|_{1+\epsilon}\right\}. \quad (33)$$

Next, we proceed to bound the second term on the right side of inequality (33) using Lemma 9. The application of Lemma 9 requires satisfying two conditions. The first condition is that the expectation of the median term $X_\tau$ is 0, which is easily fulfilled due to the symmetry of rewards. The second condition is that the $(1 + \epsilon)$-th moment of $X_\tau$ is finite, which can be verified through Lemma 8, such that

$$\mathrm{E}[|X_\tau|^{1+\epsilon}] \le rv.$$

Consequently, Lemma 9 can be employed to bound the second term on the right side of (33) by setting $C = (4v)^{\frac{1}{1+\epsilon}}$ and $v_1 = rv$. This yields that

$$\left|\sum_{i=1}^{t} \alpha_\tau X_\tau\right| \le (2C \ln(4T/\delta) + 2C^{-\epsilon} rv)\|\boldsymbol{\alpha}\|_{1+\epsilon}$$

holds with probability at least $1 - \delta/T$. Hence, the proof of Lemma 10 is concluded. $\qquad\square$

Similar to the discussion of Lemma 10, we present Lemma 11 to bound the term $\sum_{\tau=1}^{t} \beta_\tau^2 X_\tau^2$. The proof of Lemma 11 is provided in Section F.

**Lemma 11** *Let* $r = \left\lceil 16 \ln \frac{4T}{\delta} \right\rceil$, *for any* $t > 0$, *with probability at least* $1 - \delta/T$,

$$\sum_{\tau=1}^{t} \beta_\tau^2 X_\tau^2 \le C\rho\|\boldsymbol{\beta}\|_{1+\epsilon}^2$$

*where*

$$\boldsymbol{\beta} = [\beta_1, \beta_2, \ldots, \beta_t], C = (4v)^{\frac{1}{1+\epsilon}}, \rho = 2C \ln(4T/\delta) + 2C^{-\epsilon} rv.$$

By taking Lemma 10 and Lemma 11 into inequality (31), we get that

$$\|\hat{\boldsymbol{\theta}}_{t+1} - \boldsymbol{\theta}_*\|_{\mathbf{V}_{t+1}}^2 \le \|\hat{\boldsymbol{\theta}}_1 - \boldsymbol{\theta}_*\|_{\mathbf{V}_1}^2 + \sum_{\tau=1}^{t} \left(\mu(\boldsymbol{x}_\tau^\top \boldsymbol{\theta}_*) - \mu(\boldsymbol{x}_\tau^\top \hat{\boldsymbol{\theta}}_\tau)\right)^2 \|\boldsymbol{x}_\tau\|_{\mathbf{V}_\tau^{-1}}^2$$

$$- \frac{\kappa}{2}\|\boldsymbol{\alpha}\|_2^2 + 2\rho\|\boldsymbol{\alpha}\|_{1+\epsilon} + C\rho\|\boldsymbol{\beta}\|_{1+\epsilon}^2 \quad (34)$$

holds with probability at least $1 - 2\delta/T$.

Recall the upper bound of $\mu(\cdot)$ is $U$ and $\|\hat{\boldsymbol{\theta}}_1 - \boldsymbol{\theta}_*\|_{\mathbf{V}_1}^2 \le \lambda S^2$, inequality (34) can be simplified as

$$\|\hat{\boldsymbol{\theta}}_{t+1} - \boldsymbol{\theta}_*\|_{\mathbf{V}_{t+1}}^2 \le \lambda S^2 + 4U^2 \sum_{\tau=1}^{t}\|\boldsymbol{x}_\tau\|_{\mathbf{V}_\tau^{-1}}^2 - \frac{\kappa}{2}\|\boldsymbol{\alpha}\|_2^2 + 2\rho\|\boldsymbol{\alpha}\|_{1+\epsilon} + C\rho\|\boldsymbol{\beta}\|_{1+\epsilon}^2 \qquad (35)$$

Based on the Hölder inequality, we get that

$$\|\boldsymbol{\alpha}\|_{1+\epsilon} \le t^{\frac{1-\epsilon}{2(1+\epsilon)}}\|\boldsymbol{\alpha}\|_2, \ \|\boldsymbol{\beta}\|_{1+\epsilon}^2 \le t^{\frac{1-\epsilon}{1+\epsilon}}\|\boldsymbol{\beta}\|_2^2.$$

By taking these two inequalities into (35) and recalling that $\beta_\tau = \|\boldsymbol{x}_\tau\|_{\mathbf{V}_\tau^{-1}}$, we get that

$$\|\hat{\boldsymbol{\theta}}_{t+1} - \boldsymbol{\theta}_*\|_{\mathbf{V}_{t+1}}^2 \le \lambda S^2 + \left(4U^2 + C\rho t^{\frac{1-\epsilon}{1+\epsilon}}\right)\sum_{\tau=1}^{t}\|\boldsymbol{x}_\tau\|_{\mathbf{V}_\tau^{-1}}^2 - \frac{\kappa}{2}\|\boldsymbol{\alpha}\|_2^2 + 2\rho t^{\frac{1-\epsilon}{2(1+\epsilon)}}\|\boldsymbol{\alpha}\|_2.$$

holds with probability at least $1 - 2\delta/T$.

According to the fact $2\sqrt{pq} \le \frac{p}{\kappa} + \kappa q, \forall p, q > 0$, if we take $p = 4\rho^2 t^{\frac{1-\epsilon}{1+\epsilon}}, q = \|\boldsymbol{\alpha}\|_2^2$, we get that

$$\|\hat{\boldsymbol{\theta}}_{t+1} - \boldsymbol{\theta}_*\|_{\mathbf{V}_{t+1}}^2 \le \left(4U^2 + C\rho t^{\frac{1-\epsilon}{1+\epsilon}}\right)\sum_{\tau=1}^{t}\|\boldsymbol{x}_\tau\|_{\mathbf{V}_\tau^{-1}}^2 + \lambda S^2 + \frac{2\rho^2}{\kappa}t^{\frac{1-\epsilon}{1+\epsilon}}.$$

holds with probability at least $1 - 2\delta/T$. Then, take an union bound over all rounds and , we have that with probability at least $1 - 2\delta$, for any $t > 0$,

$$\|\hat{\boldsymbol{\theta}}_{t+1} - \boldsymbol{\theta}_*\|_{\mathbf{V}_{t+1}}^2 \le \left(4U^2 + C\rho t^{\frac{1-\epsilon}{1+\epsilon}}\right)\frac{4d}{\kappa}\ln\left(1 + \frac{\kappa t}{2\lambda d}\right) + \lambda S^2 + \frac{2\rho^2}{\kappa}t^{\frac{1-\epsilon}{1+\epsilon}}.$$

The proof of Theorem 3 is finished.

# D  Proof of Theorem 4

Since CRMM plays total $T_0$ rounds with $T_0 = \lfloor T/r \rfloor$ and $r = \lceil 16\ln\frac{4T}{\delta}\rceil$, we bound the sum of $\gamma_t$ from $t = 1$ to $T_0$ first, such that

$$\sum_{t=1}^{T_0}\gamma_t \le \left(\frac{16U^2 d}{\kappa}\ln\left(1 + \frac{\kappa T_0}{2\lambda d}\right) + \lambda S^2\right)T_0 + \left(\frac{2\rho^2}{\kappa}T_0 + \frac{4C\rho d}{\kappa}\ln\left(1 + \frac{\kappa T_0}{2\lambda d}\right)\right)\sum_{t=1}^{T_0}t^{\frac{1-\epsilon}{1+\epsilon}}$$

$$\le \left(\frac{16U^2 d}{\kappa}\ln\left(1 + \frac{\kappa T_0}{2\lambda d}\right) + \lambda S^2\right)T_0 + \left(\frac{2\rho^2}{\kappa} + \frac{4C\rho d}{\kappa}\ln\left(1 + \frac{\kappa T_0}{2\lambda d}\right)\right)T_0^{\frac{2}{1+\epsilon}}.$$

The second inequality holds due to the fact $\sum_{t=1}^{T_0}t^{\frac{1-\epsilon}{1+\epsilon}} \le \int_0^{T_0}x^{\frac{1-\epsilon}{1+\epsilon}}\mathrm{d}x \le T_0^{\frac{2}{1+\epsilon}}$. Taking above result into Lemma 7, we can easily get that

$$R(T_0) \le 16LUd\kappa^{-1}\ln\left(1 + \frac{\kappa T_0}{2\lambda d}\right)T_0^{\frac{1}{2}} + 4LS\kappa^{-\frac{1}{2}}\left(\lambda d\ln\left(1 + \frac{\kappa T_0}{2\lambda d}\right)\right)^{\frac{1}{2}}T_0^{\frac{1}{2}}$$

$$+ 8L\rho\kappa^{-1}\left(d\ln\left(1 + \frac{\kappa T_0}{2\lambda d}\right)\right)^{\frac{1}{2}}T_0^{\frac{1}{1+\epsilon}} + 8Ld\kappa^{-1}C\rho^{\frac{1}{2}}\ln\left(1 + \frac{\kappa T_0}{2\lambda d}\right)T_0^{\frac{1}{1+\epsilon}}.$$

Taking $R(T) = rR(T_0)$ shows that the regret of CRMM can be bounded as

$$R(T) \le 64LUd\kappa^{-1}\ln\left(1 + \frac{\kappa T}{2\lambda d}\right)\left(\ln\frac{4T}{\delta}\right)^{\frac{1}{2}}T^{\frac{1}{2}}$$

$$+ 16LS\kappa^{-\frac{1}{2}}\left(\lambda d\ln\left(1 + \frac{\kappa T}{2\lambda d}\right)\ln\frac{4T}{\delta}\right)^{\frac{1}{2}}T^{\frac{1}{2}}$$

$$+ 32L\rho\kappa^{-1}\left(d\ln\left(1 + \frac{\kappa T}{2\lambda d}\right)\right)^{\frac{1}{2}}\left(\ln\frac{4T}{\delta}\right)^{\frac{\epsilon}{1+\epsilon}}T^{\frac{1}{1+\epsilon}}$$

$$+ 32Ld\kappa^{-1}C\rho^{\frac{1}{2}}\ln\left(1 + \frac{\kappa T}{2\lambda d}\right)\left(\ln\frac{4T}{\delta}\right)^{\frac{\epsilon}{1+\epsilon}}T^{\frac{1}{1+\epsilon}}$$

$$= O\left(d(\log T)^{\frac{3}{2} + \frac{\epsilon}{1+\epsilon}}T^{\frac{1}{1+\epsilon}}\right).$$

The proof of Theorem 4 is finished.

# E  Proof of Lemma 9

Let $Z_i = X_i \mathbb{I}_{|\alpha_i X_i| \leq C\|\boldsymbol{\alpha}\|_{1+\epsilon}}$. Based on the triangle inequality, we obtain that

$$\left| \sum_{i=1}^{n} \alpha_i Z_i \right| \leq \left| \sum_{i=1}^{n} \alpha_i Z_i - \mathrm{E}\left[\alpha_i Z_i | \mathcal{F}_{i-1}\right] \right| + \left| \sum_{i=1}^{n} \mathrm{E}\left[\alpha_i Z_i | \mathcal{F}_{i-1}\right] \right|. \tag{36}$$

Utilizing Bernstein's inequality [Seldin *et al.*, 2012, Lemma 11] for the first term in the right side of obove equation shows that with probability at least $1 - \delta$, we have

$$\left| \sum_{i=1}^{n} \alpha_i Z_i - \mathrm{E}\left[\alpha_i Z_i | \mathcal{F}_{i-1}\right] \right| \leq 2C\|\boldsymbol{\alpha}\|_{1+\epsilon} \ln(2/\delta) + \frac{1}{2C\|\boldsymbol{\alpha}\|_{1+\epsilon}} \sum_{i=1}^{n} \mathrm{Var}[\alpha_i Z_i | \mathcal{F}_{i-1}].$$

The variance of $Z_i$ can be relaxed as follows,

$$\begin{aligned}
\sum_{i=1}^{n} \mathrm{Var}[\alpha_i Z_i | \mathcal{F}_{i-1}] &= \sum_{i=1}^{n} \mathrm{E}\left[ (\alpha_i Z_i - \mathrm{E}[\alpha_i Z_i | \mathcal{F}_{i-1}])^2 | \mathcal{F}_{i-1} \right] \\
&\leq \sum_{i=1}^{n} \mathrm{E}\left[ (\alpha_i Z_i)^2 | \mathcal{F}_{i-1} \right] \leq v C^{1-\epsilon} \|\boldsymbol{\alpha}\|_{1+\epsilon}^2.
\end{aligned}$$

Thus, we get that

$$\left| \sum_{i=1}^{n} \alpha_i Z_i - \mathrm{E}\left[\alpha_i Z_i | \mathcal{F}_{i-1}\right] \right| \leq (2C \ln(2/\delta) + v C^{-\epsilon}) \|\boldsymbol{\alpha}\|_{1+\epsilon} \tag{37}$$

holds with probability at least $1 - \delta$.

According to the conditions $\mathrm{E}[X_i | \mathcal{F}_{i-1}] = 0$ and $\mathrm{E}[|X_i|^{1+\epsilon} | \mathcal{F}_{i-1}] \leq v$ for $i = 1, 2, \ldots, n$, we can easily obtain that

$$\begin{aligned}
\left| \sum_{i=1}^{n} \mathrm{E}\left[\alpha_i Z_i | \mathcal{F}_{i-1}\right] \right| &= \left| \sum_{i=1}^{n} \mathrm{E}\left[ \alpha_i X_i \mathbb{I}_{|\alpha_i X_i| \leq C\|\boldsymbol{\alpha}\|_{1+\epsilon}} | \mathcal{F}_{i-1} \right] \right| \\
&\leq \sum_{i=1}^{n} \mathrm{E}\left[ |\alpha_i X_i| \mathbb{I}_{|\alpha_i X_i| > C\|\boldsymbol{\alpha}\|_{1+\epsilon}} | \mathcal{F}_{i-1} \right] \\
&\leq \sum_{i=1}^{n} \left( \mathrm{E}\left[ |\alpha_i X_i|^{1+\epsilon} | \mathcal{F}_{i-1} \right] \right)^{\frac{1}{1+\epsilon}} \mathrm{Pr}\left\{ |\alpha_i X_i| > C\|\boldsymbol{\alpha}\|_{1+\epsilon} \right\}^{\frac{\epsilon}{1+\epsilon}} \\
&= v C^{-\epsilon} \|\boldsymbol{\alpha}\|_{1+\epsilon}.
\end{aligned} \tag{38}$$

Taking (37) and (38) into (36) finishes the proof.

# F  Proof of Lemma 11

We first provide the following lemma to help with the proof of Lemma 11.

**Lemma 12** *Let $X_1, \ldots, X_n$ be random variables with bounded moments $\mathrm{E}[|X_i|^{1+\epsilon} | \mathcal{F}_{i-1}] \leq v_1$, where $\mathcal{F}_{i-1} \triangleq \{X_1, \ldots, X_{i-1}\}$ is a $\sigma$-filtration and $\mathcal{F}_0 = \emptyset$. For the fixed parameters $\beta_1, \beta_2, \ldots, \beta_n \in \mathbb{R}$ and $C > 0$, with probability at least $1 - \delta$, we have that*

$$\sum_{i=1}^{n} \beta_i^2 X_i^2 \mathbb{I}_{\beta_i^2 X_i^2 \leq C^2 \|\boldsymbol{\beta}\|_{1+\epsilon}^2} \leq \xi \|\boldsymbol{\beta}\|_{1+\epsilon}^2,$$

*where*

$$\boldsymbol{\beta} = [\beta_1, \beta_2, \ldots, \beta_n], \xi = 2C^2 \ln(2/\delta) + 2v_1 C^{1-\epsilon}.$$

**Proof.** Let $Z_i^2 = X_i^2 \mathbb{I}_{\beta_i^2 X_i^2 \leq C^2 \|\boldsymbol{\beta}\|_{1+\epsilon}^2}$. The triangle inequality shows that

$$\sum_{i=1}^n \beta_i^2 Z_i^2 \leq \left| \sum_{i=1}^n \beta_i^2 Z_i^2 - \mathrm{E}\left[\beta_i^2 Z_i^2 | \mathcal{F}_{i-1}\right] \right| + \left| \sum_{i=1}^n \mathrm{E}\left[\beta_i^2 Z_i^2 | \mathcal{F}_{i-1}\right] \right|. \tag{39}$$

Taking use of the Bernstein's inequality [Seldin *et al.*, 2012, Lemma 11] tells that

$$\left| \sum_{i=1}^n \beta_i^2 Z_i^2 - \mathrm{E}\left[\beta_i^2 Z_i^2 | \mathcal{F}_{i-1}\right] \right| \leq 2C^2 \|\boldsymbol{\beta}\|_{1+\epsilon}^2 \ln(2/\delta) + \frac{1}{2C^2 \|\boldsymbol{\beta}\|_{1+\epsilon}^2} \sum_{i=1}^n \mathrm{Var}[\beta_i^2 Z_i^2 | \mathcal{F}_{i-1}]$$

holds with probability at least $1 - \delta$. The variance of $\beta_i^2 Z_i^2$ can be relaxed as

$$\sum_{i=1}^n \mathrm{E}\left[(\beta_i^2 Z_i^2 - \mathrm{E}[\beta_i^2 Z_i^2])^2 | \mathcal{F}_{i-1}\right] \leq \sum_{i=1}^n \mathrm{E}\left[(\beta_i Z_i)^4 | \mathcal{F}_{i-1}\right] \leq v_1 C^{3-\epsilon} \|\boldsymbol{\beta}\|_{1+\epsilon}^4.$$

Thus, we get that

$$\left| \sum_{i=1}^n \beta_i^2 Z_i^2 - \mathrm{E}\left[\beta_i^2 Z_i^2 | \mathcal{F}_{i-1}\right] \right| \leq 2C^2 \|\boldsymbol{\beta}\|_{1+\epsilon}^2 \ln(2/\delta) + v_1 C^{1-\epsilon} \|\boldsymbol{\beta}\|_{1+\epsilon}^2. \tag{40}$$

Considering that $\mathrm{E}[|X_i|^{1+\epsilon} | \mathcal{F}_{i-1}] \leq v_1$, $i = 1, 2, \ldots, n$, it is easy to verify that

$$\sum_{i=1}^n \mathrm{E}\left[\beta_i^2 Z_i^2 | \mathcal{F}_{i-1}\right] \leq v_1 C^{1-\epsilon} \|\boldsymbol{\beta}\|_{1+\epsilon}^2. \tag{41}$$

Taking (40) and (41) into (39) finishes the proof of Lemma 12. $\qquad\square$

Now, we are ready to prove Lemma 11. Through the full probability formula [Mendenhall *et al.*, 2012], we have that

$$\begin{aligned}
\Pr\left\{ \sum_{\tau=1}^t \beta_\tau^2 X_\tau^2 > C\rho \|\boldsymbol{\beta}\|_{1+\epsilon}^2 \right\} \leq & \Pr\left\{ \sum_{\tau=1}^t \beta_\tau^2 X_\tau^2 \mathbb{I}_{\beta_\tau^2 X_\tau^2 \leq C^2 \|\boldsymbol{\beta}\|_{1+\epsilon}^2} > C\rho \|\boldsymbol{\beta}\|_{1+\epsilon}^2 \right\} \\
& + \sum_{\tau=1}^t \Pr\left\{ |\beta_\tau X_\tau| > C\|\boldsymbol{\beta}\|_{1+\epsilon} \right\}.
\end{aligned} \tag{42}$$

We first analyze the second term on the right side of above inequality. Recall that CRMM observes $r$ rewards $\{y_t^1, \ldots, y_t^r\}$ at round $t$, and for the sake of representation, we denote the difference between $y_t^i$ and $\mu(\boldsymbol{x}_t^\top \boldsymbol{\theta}_*)$ as $X_t^i$, such that $X_t^i = y_t^i - \mu(\boldsymbol{x}_t^\top \boldsymbol{\theta}_*)$. Through Markov's inequality and the heavy-tailed condition $\mathrm{E}[|X_\tau|^{1+\epsilon}] \leq v$, we have that

$$\Pr\left\{ |\beta_\tau X_\tau^i| > C\|\boldsymbol{\beta}\|_{1+\epsilon} \right\} \leq \frac{|\beta_\tau|^{1+\epsilon} v}{C^{1+\epsilon} \|\boldsymbol{\beta}\|_{1+\epsilon}^{1+\epsilon}}.$$

Let $C = (4v)^{\frac{1}{1+\epsilon}}$, then we have

$$\Pr\left\{ |\beta_\tau X_\tau^i| > C\|\boldsymbol{\beta}\|_{1+\epsilon} \right\} \leq \frac{1}{4}.$$

Define the random variables

$$B_i = \mathbb{I}_{\beta_\tau X_\tau^i > C\|\boldsymbol{\beta}\|_{1+\epsilon}},$$

thus $p_i = \Pr\{B_i = 1\} \leq \frac{1}{4}$. According to the Azuma-Hoeffing's inequality [Azuma, 1967], we have that

$$\begin{aligned}
\Pr\left\{ \sum_{i=1}^r B_j \geq \frac{r}{2} \right\} \leq & \Pr\left\{ \sum_{i=1}^r B_i - p_i \geq \frac{r}{4} \right\} \\
\leq & e^{-r/8} \leq \frac{\delta}{4T^2}
\end{aligned}$$

for $r = \lceil 16 \ln \frac{4T}{\delta} \rceil$. The inequality $\sum_{i=1}^{r} B_i \geq \frac{r}{2}$ means more than half of the terms $\{B_i\}_{i=1}^{r}$ is true. Thus, the median term $\beta_\tau X_\tau$ satisfies

$$\beta_\tau X_\tau > C\|\boldsymbol{\beta}\|_{1+\epsilon}$$

with probability at most $\frac{\delta}{4T^2}$. A similar argument shows that

$$\beta_\tau X_\tau < -C\|\boldsymbol{\beta}\|_{1+\epsilon}$$

holds with probability at most $\frac{\delta}{4T^2}$. Therefore, we have

$$\Pr\left\{ |\beta_\tau X_\tau| > C\|\boldsymbol{\beta}\|_{1+\epsilon} \right\} \leq \frac{\delta}{2T^2}.$$

Take it into (42), we get that

$$\Pr\left\{ \sum_{\tau=1}^{t} \beta_\tau^2 X_\tau^2 > C\rho\|\boldsymbol{\beta}\|_{1+\epsilon} \right\} \leq \frac{\delta}{2T} + \Pr\left\{ \sum_{\tau=1}^{t} \beta_\tau^2 X_\tau^2 \mathbb{I}_{\beta_\tau^2 X_\tau^2 \leq C^2\|\boldsymbol{\beta}\|_{1+\epsilon}^2} > C\rho\|\boldsymbol{\beta}\|_{1+\epsilon}^2 \right\}.$$

We obtain that the $(1+\epsilon)$-th moment of $X_\tau$ is $rv$ by Lemma 8, thus Lemma 12 can be taken to bound the second term on the right side of above inequality, such that

$$\Pr\left\{ \sum_{\tau=1}^{t} \beta_\tau^2 X_\tau^2 \mathbb{I}_{\beta_\tau^2 X_\tau^2 \leq C^2\|\boldsymbol{\beta}\|_{1+\epsilon}^2} > C\rho\|\boldsymbol{\beta}\|_{1+\epsilon}^2 \right\} \leq \frac{\delta}{2T}$$

with $\rho = 2C\ln(4T/\delta) + 2C^{-\epsilon}rv$. Thus, we get that

$$\sum_{\tau=1}^{t} \beta_\tau^2 X_\tau^2 \leq (2C^2\ln(4T/\delta) + 2rvC^{1-\epsilon})\|\boldsymbol{\beta}\|_{1+\epsilon}^2$$

holds with probability at least $1 - \delta/T$.