# OpenReview forum: "Efficient Algorithms for Generalized Linear Bandits with Heavy-tailed Rewards"
_NeurIPS.cc/2023/Conference — NeurIPS 2023 poster_

### Official Review · Reviewer_jQHU · 2023-06-15

**Soundness:** 3 good
**Presentation:** 3 good
**Contribution:** 3 good
**Rating:** 6
**Confidence:** 4

**Summary:**

In this paper, the authors study the problem of generalized linear bandits with heavy tailed rewards. They propose two algorithms based on truncation and mean of medians. The algorithms both achieve near optimal regret bound of $\tilde{O}(dT^{\frac{1}{1+\epsilon}})$. These regret bounds improve upon previous results in the sense of regret bound or computational complexity. Finally, the authors run simulations to support their claims.

**Strengths:**

1. The problem of generalized linear bandits with heavy-tailed rewards is well-motivated.
2. Both algorithms achieve near optimal regret bounds, the proof looks correct to me.
3. The writing is clear, the related work part is really helpful.

**Weaknesses:**

1. According to the introduction of previous methods for generalized linear bandits and bandits with heavy-tailed rewards, the algorithms are direct combinations of previous techniques. Therefore the algorithmic novelty is limited.

**Questions:**

1. For CRMM, since the assumption of symmetric distribution is used to guarantee that the median is 0, I am curious about whether it is possible to replace the condition with the median of the distribution equals 0.

**Limitations:**

Yes.

---

> ### Author Rebuttal · Authors · 2023-08-07
>
> Thank you sincerely for your time and effort in reviewing our work. We have carefully considered your concerns and our responses are provided as follows.
>
> ----
>
> **Weaknesses: According to the introduction of previous methods for generalized linear bandits and bandits with heavy-tailed rewards, the algorithms are direct combinations of previous techniques. Therefore the algorithmic novelty is limited.**
>
> In fact, all existing heavy-tailed bandit algorithms are combinations of bandit algorithms' fundamental framework and heavy-tailed strategies (truncation and mean of medians) (Bubeck et al., 2013; Medina and Yang, 2016; Shao et al., 2018; Xue et al., 2020). The challenge is to find an appropriate approach to apply the heavy-tailed strategies for the problems at hand, and provide rigorous analysis for the proposed algorithms.
>
> Our proposed algorithms differ from existing algorithms (Shao et al., 2018; Xue et al., 2020) in two main aspects. First, our work integrates heavy-tailed strategies into the Online Newton Step (ONS) method, whereas existing research has applied heavy-tailed strategies to Least Square Estimation (LSE), resulting in fundamentally different analytical techniques. Second, our algorithms not only achieve nearly optimal regret bounds but also offer significant improvements in efficiency compared to existing heavy-tailed algorithms (Shao et al., 2018; Xue et al., 2020). Specifically, CRTM reduces the computational complexity from $O(T^2)$ to $O(T)$ when compared to existing truncation-based algorithms (Shao et al., 2018; Xue et al., 2020). CRMM reduces the number of estimators required per round from $O(\log T)$ to only $1$ when compared to existing median-of-means-based algorithms (Shao et al., 2018; Xue et al., 2020). To attain such improvements, we have developed novel approaches for applying heavy-tailed strategies and introduced new analytical techniques. We provide a detailed introduction as follows.
>
> **Truncation-based algorithm:** CRTM differs from existing algorithms TOFU (Shao et al., 2018) and BTC (Xue et al., 2020) in the truncated terms. Both TOFU and BTC have to store the historical rewards and truncate all these rewards at each epoch, resulting in a computational complexity of $O(T^2)$. In contrast, CRTM achieves online learning by truncating only the reward of current round, whose computational complexity is $O(T)$. The main innovation in CRTM's analysis is the adoption of inductive method. Please refer to the **response for reviewer QvG4** for more details.
>
> **Mean-of-medians-based algorithm:** CRMM differs from MENU (Shao et al., 2018) and BMM (Xue et al., 2020) in the order of the "mean" and "medians" operations. MENU and BMM first calculate $O(\log T)$ "means" using multiple historical reward sequences, and then takes the median of these "means". CRMM reverses the order of these two operations to reduce the number of "means", as the calculation of "means" is time-consuming. Specifically, CRMM first takes the median of multiple rewards and then calculates a single "mean" using the median rewards, which reduces the number of estimators required per round from $O(\log T)$ to only $1$. The analysis has been adapted to prove such an exchange can achieve the nearly optimal regret bound. Two important properties of the median are utilized in our proof. The first property is that scaling a set of variables does not alter the index of the median term, which is employed in the proof of Lemma 10 (Lines 144-153). The second property is the upper bound of the median's $(1+\epsilon)$-th moment, as established in Lemma 8.
>
> ----
>
> **Questions: For CRMM, since the assumption of symmetric distribution is used to guarantee that the median is 0, I am curious about whether it is possible to replace the condition with the median of the distribution equals 0.**
>
>
> Yes, replacing the condition with the median of the distribution being equal to $0$ is sufficient to conduct the nearly optimal regret bound. The symmetric distribution is utilized in the proof of Lemma 10, and the details can be found in Lines 156-158 of the supplementary material, where we mention that the symmetric distribution is employed to ensure that the median is 0. We will provide a more comprehensive discussion of this assumption in the revised paper.

---

### Official Review · Reviewer_7N78 · 2023-07-02

**Soundness:** 3 good
**Presentation:** 4 excellent
**Contribution:** 3 good
**Rating:** 6
**Confidence:** 3

**Summary:**

This paper considers Generalized Linear Bandit (GLB) with heavy-tailed rewards, i.e., the rewards $y_t=\mu(\langle x_t,\theta^\ast\rangle)+\eta_t$ only allows a bounded $(1+\epsilon)$-order moment. By utilizing the truncation and the mean of medians technique (both for handling heavy-tailed r.v.s), the authors establish efficient algorithms for GLBs with infinite arms and heavy-tailed rewards.

**Strengths:**

1. Compared to previous works on LB with heavy-tailed rewards, this paper considers Generalized LB instead of standard Stochastic LB, where the function $\mu$ can be any Lipschitz and uniformly bounded function.
2. This paper has a lower computational complexity: previous truncation-based works achieving $\sqrt T$-style regret require performing an iteration over all previous observations every time they update the confidence set, which costs $\mathcal O(T)$ time. In contrast, by utilizing the ONS step used in the $\text{OL}^2\text{M}$ algorithm, this algorithm only needs $\mathcal O(d^2)$ time for each round.
3. The presentation is pretty clear, and the algorithms are easy to understand.
4. The results are supported by various numerical illustrations.

**Weaknesses:**

I am not sure what is the main technical contribution of this paper. For example, the CRTM algorithm looks pretty like equipping the $\text{OL}^2\text{M}$ algorithm by Zhang et al. (2016) with the previous truncation techniques by Shao et al. (2018) or Xue et al. (2020). Without any part in the main text devoted to explaining the technical hardness of showing Theorems 1 & 2, it is hard to evaluate the contribution of this paper. The same also holds for the mean-of-medians method CRMM.

**Questions:**

See Weaknesses. I am willing to increase my score if the authors can address it convincingly.

---

> ### Author Rebuttal · Authors · 2023-08-07
>
> Thanks sincerely for your time and effort in reviewing our work.  We have carefully considered your concerns, and are very happy to respond more questions during the rolling discussion.
>
> ----
>
> **Q1:The CRTM algorithm looks pretty like equipping the OL$^2$M algorithm by Zhang et al. (2016) with the previous truncation techniques by Shao et al. (2018) or Xue et al. (2020).  The same also holds for the mean-of-medians method CRMM.**
>
> To deal with the heavy-tailed GLB problem, it is essential to incorporate the GLB algorithm's fundamental technique (Online Newton Step method) and heavy-tailed strategies (truncation and mean of medians). However, our algorithms not only attain the nearly optimal regret bounds but also offer significant improvements in efficiency compared with existing algorithms (Shao et al., 2018; Xue et al., 2020). Specifically, CRTM reduces the computational complexity from $O(T^2)$ to $O(T)$, and CRMM reduces the number of estimators required per round from $O(\log T)$ to only $1$. To achieve such improvements, we have developed novel approaches for applying heavy-tailed strategies and conducted new analytical techniques. We give a detailed introduction as follows.
>
> ----
> **Q2: Technical contribution of the truncation-based method CRTM.**
>
> Compared with existing truncation-based algorithm TOFU (Shao et al., 2018) and BTC (Xue et al., 2020), CRTM differs from them in two aspects. First, both TOFU and BTC have to store the historical rewards and truncate all these rewards at each epoch, resulting in a computational complexity of $O(T^2)$. In contrast, CRTM achieves online learning by truncating only the reward of current round, whose computational complexity is $O(T)$. Second, TOFU and BTC are designed for SLB model and calculate the estimator via Least Square Estimation (LSE) , CRTM is designed for the GLB model and updates the estimator using the Online Newton Step (ONS) method, which makes the analytical techniques different.
>
> The main innovation in CRTM's analysis is the adoption of inductive method. We first display the key differences in the analysis between TOFU and CRTM. For TOFU employing LSE, the confidence region for the inherent parameter $\theta_\*$ is
> \begin{equation}
> \lVert\hat{\theta}\_{t+1}^{LSE}-\theta\_\*\rVert_{\widetilde{V}\_{t+1}} \leq  \lVert \underbrace{\widetilde{V}\_{t+1}^{-\frac{1}{2}} A\_t\(Y\_t-A\_t^\top\theta\_\*\)}\_{A}  \rVert\_2+\lVert\theta\_\*\rVert\_{\widetilde{V}\_{t+1}^{-1}}
> \end{equation}
> where $A\_t=[x\_1,x\_2,\ldots,x\_t]\in\mathbb{R}^{d\times t}$ is the matrix composed of selected arm vectors, $\widetilde{V}\_{t+1}=A\_tA\_t^\top+I_d$ and $Y\_t=[y\_1,y\_2,\ldots,y\_t]\in\mathbb{R}^{t\times 1}$ is the historical reward vector. For CRTM employing the ONS method, the confidence region for the inherent parameter $\theta\_\*$ is
> \begin{equation}
> \lVert\hat{\theta}\_{t+1}^{ONS}-\theta\_\*\rVert\_{V\_{t+1}}^2\leq\underbrace{\sum\_{\tau=1}^t 2x\_\tau^\top\left\(\hat{\theta}\_\tau^{ONS}-\theta\_\*\right\) \left\(y\_\tau-\mu\(x\_\tau^\top\theta\_\*\)\right\)}\_{B}+\sum\_{\tau=1}^t \lVert x_\tau\rVert\_{V\_\tau^{-1}}^2 \left\(y\_\tau-\mu\(x\_\tau^\top\theta\_\*\)\right\)^2+O\(\log t\).
> \end{equation}
> Although both above two equations include the linear combination of historical rewards ($A$ and $B$), there exists essential difference between these two terms. Specifically, $A$'s coefficients $\widetilde{V}_{t+1}^{-\frac{1}{2}}A_t$ are known and the agent can truncate the scaled rewards to reduce the impact of extreme noises. However, $B$'s coefficients $\\{x\_\tau^\top\(\hat{\theta}\_\tau^{ONS}-\theta\_*\)\\}\_{\tau=1}^t$ are unknown, which makes the existing technique of TOFU invalid.
>
> To address this issue, we adopt the inductive method. The first step is to relax term $B$ as follows:
> \begin{equation}
> \|x\_\tau^\top\(\hat{\theta}^{ONS}\_\tau-\theta\_\*\)y\_\tau\|\leq\lVert\hat{\theta}\_\tau^{ONS}-\theta\_\*\rVert\_{V\_\tau}\cdot\lVert x\_\tau\rVert\_{V\_\tau^{-1}}\cdot|y\_\tau|.
> \end{equation}
> For $\tau=1$, it is evident that $\lVert\hat{\theta}\_1^{ONS}-\theta\_\*\rVert\_{V\_1}^2\leq \gamma$. Then, we assume $\lVert\hat{\theta}\_\tau^{ONS}-\theta\_\*\rVert\_{V\_\tau}^2\leq \gamma$ for $\tau=1,2,\ldots, t$, which replaces the unknown parameters $\lVert\hat{\theta}\_\tau^{ONS}-\theta\_\*\rVert\_{V\_\tau}$ with $\gamma^{1/2}$. Since $\lVert x\_\tau\rVert\_{V\_\tau^{-1}}$ is known, CRTM can truncate the term $\lVert x\_\tau\rVert\_{V\_\tau^{-1}}\|y\_\tau|\$ to reduce the impact of extreme values. A delicate analysis of this approach for applying the truncated strategy demonstrates that $\lVert\hat{\theta}\_{t+1}^{ONS}-\theta\_\*\rVert\_{V\_{t+1}}^2\leq\gamma$, which concludes the proof of the confidence region. Further details can be found in Lemma 5 of the supplementary material.
>
> ----
>
> **Q3: Technical contribution of the mean-of-medians method CRMM.**
>
> Compared with existing median of means algorithms MENU (Shao et al., 2018) and BMM (Xue et al., 2020), CRMM differs from them in two aspects. The first one is the estimator as we introduced in Q2. The second one is the order of "mean" and  "medians'' operations. MENU and BMM first calculate $O(\log T)$ "means'' using multiple historical reward sequences, and then takes the median of these "means''. CRMM reverses the order of these two operations to reduce the number of "means'' to $1$, as the calculation of "means'' is time-consuming. Specifically, CRMM first takes the median of multiple rewards and then calculates a single "mean'' using the median reward.
>
> The analysis has been adapted to prove that such an exchange can achieve a near-optimal regret bound. Two important properties of the median are utilized in our proof. The first property is that scaling a set of variables does not alter the index of the median term, which is employed in the proof of Lemma 10 (Lines 144-153). The second property is the upper bound of the median's $(1+\epsilon)$-th moment, as established in Lemma 8.

---

> > ### Comment · Reviewer_7N78 · 2023-08-21
> >
> > Thank you for your clarification. I'm glad to recommend an acceptance. It would be nice if the authors can include more discussions on these differences in the main text of the final version.

---

> > > ### Author Response · Authors · 2023-08-21
> > >
> > > Dear Reviewer 7N78,
> > >
> > > Thank you very much for your kind reply! We will highlight the technical contributions in the main text of the final version.
> > >
> > > Best,
> > > Authors

---

### Official Review · Reviewer_QvG4 · 2023-07-02

**Soundness:** 3 good
**Presentation:** 4 excellent
**Contribution:** 2 fair
**Rating:** 6
**Confidence:** 4

**Summary:**

This paper studies the problem of generalized linear bandits with heavy-tail rewards. Due to the heavy-tailedness, methods and algorithms for linear bandits with sub-gaussian rewards cannot be directly applied. To handle such issues, existing works have developed certain strategies, two of which are the truncation strategy and the mean-of-medians strategy. To this end, this paper proposes two novel algorithms, CRTM and CRMM, which utilize the aforementioned strategies and achieve sublinear regret bounds. CRTM reduces the computational complexity of previously best known truncation-based methods, while CRMM reduces the number of estimators required. Furthermore, CRTM does not require the reward to have symmetric distribution, contrasting existing works. Experimental results demonstrate the low regret and computational complexity of the proposed methods.


**Strengths:**


**1. Clarity**
The paper has a clear presentation. The GLB bandit problem and heavy-tailedness are rigorously defined.

**2. Meaningful results for an important problem**
Heavy-tailedness is an important aspect due to its board application in many real-world scenarios. This paper proposed computationally efficient near-optimal algorithms for such a setting, which is a meaningful result.

**3. Experiments**
The experimental study covers the regret bound as well as the computational complexity.


**Weaknesses:**

There is no major technical weakness/flaws in this paper, as far as I can tell.

However, I do challenge the technical contribution of this paper: in terms of the theoretical analysis, does this paper improves over existing results or does this paper develop novel analytical techniques to facilitate the analysis? For example, in line 209, this paper mentions that the theoretical analysis is fundamentally different from that of TOFU/BTC. However, it seems to me that this is not elaborated in the later part of the paper, so I have to assume that the technical contribution is incremental compared to existing works.  Therefore, I am concerned about the technical contribution of this paper.


**Questions:**

In the theoretical proof, what are the major differences from previous works? For example, is there any non-trivial ideas applied?
More details in  the Weakness section.


**Limitations:**

Yes.

---

> ### Author Rebuttal · Authors · 2023-08-07
>
> Thank you sincerely for your time and effort in reviewing our work.  We have carefully considered your concerns, and are very happy to respond more questions during the rolling discussion.
>
> ----
>
> **Questions: In the theoretical proof, what are the major differences from previous works TOFU/BTC? For example, is there any non-trivial ideas applied?**
>
> We first highlight the key differences in the analysis between TOFU and CRTM. For TOFU, which employs the least square estimation (LSE), the confidence region for the inherent parameter $\theta_*$ is given by
> \begin{equation}
> \lVert\hat{\theta}\_{t+1}^{LSE}-\theta\_\*\rVert_{\widetilde{V}\_{t+1}} \leq  \lVert \underbrace{\widetilde{V}\_{t+1}^{-\frac{1}{2}} A\_t\(Y\_t-A\_t^\top\theta\_\*\)}\_{A}  \rVert\_2+\lVert\theta\_\*\rVert\_{\widetilde{V}\_{t+1}^{-1}}
> \end{equation}
> where $A\_t=[x\_1,x\_2,\ldots,x\_t]\in\mathbb{R}^{d\times t}$ is the matrix composed of selected arm vectors, $\widetilde{V}\_{t+1}=A\_tA\_t^\top+I_d$ and $Y\_t=[y\_1,y\_2,\ldots,y\_t]\in\mathbb{R}^{t\times 1}$ is the historical reward vector. For CRTM, which employs the Online Newton Step (ONS) method, the confidence region for the inherent parameter $\theta\_*$ is
> \begin{equation}
> \lVert\hat{\theta}\_{t+1}^{ONS}-\theta\_\*\rVert\_{V\_{t+1}}^2\leq\underbrace{\sum\_{\tau=1}^t 2x\_\tau^\top\left\(\hat{\theta}\_\tau^{ONS}-\theta\_\*\right\) \left\(y\_\tau-\mu\(x\_\tau^\top\theta\_\*\)\right\)}\_{B}+\sum\_{\tau=1}^t \lVert x_\tau\rVert\_{V\_\tau^{-1}}^2 \left\(y\_\tau-\mu\(x\_\tau^\top\theta\_\*\)\right\)^2+O\(\log t\).
> \end{equation}
>
> Although both above two equations include the linear combination of historical rewards ($A$ and $B$), there exists essential differences between these two terms. Specifically, $A$'s coefficients $\widetilde{V}_{t+1}^{-\frac{1}{2}}A_t$ are known and the agent can truncate the scaled rewards to reduce the impact of extreme noises. However, $B$'s coefficient $\\{x\_\tau^\top\(\hat{\theta}\_\tau^{ONS}-\theta\_*\)\\}\_{\tau=1}\^t$ are unknown, which makes the existing technique of TOFU invalid.
>
> To address this issue, our main contribution in analytical technique is the adoption of inductive method. We first relax term $B$ as follows:
> \begin{equation}
> \|x\_\tau^\top\(\hat{\theta}^{ONS}\_\tau-\theta\_\*\)y\_\tau\|\leq\lVert\hat{\theta}\_\tau^{ONS}-\theta\_\*\rVert\_{V\_\tau}\cdot\lVert x\_\tau\rVert\_{V\_\tau^{-1}}\cdot|y\_\tau|.
> \end{equation}
>
> For $\tau=1$, it is evident that $\lVert\hat{\theta}\_1^{ONS}-\theta\_\*\rVert\_{V\_1}^2\leq \gamma$. Then, we assume $\lVert\hat{\theta}\_\tau^{ONS}-\theta\_\*\rVert\_{V\_\tau}^2\leq \gamma$ for $\tau=1,2,\ldots, t$, which replaces the unknown parameters $\lVert\hat{\theta}\_\tau^{ONS}-\theta\_\*\rVert\_{V\_\tau}$ with $\gamma^{1/2}$. Since $\lVert x\_\tau\rVert\_{V\_\tau^{-1}}$ is known, CRTM can truncate the term $\lVert x\_\tau\rVert\_{V\_\tau^{-1}}\|y\_\tau|\$ to reduce the impact of extreme values. A delicate analysis of this approach for applying the truncated strategy demonstrates that $\lVert\hat{\theta}\_{t+1}^{ONS}-\theta\_\*\rVert\_{V\_{t+1}}^2\leq\gamma$, which concludes the proof of the confidence region. Further details can be found in Lemma 5 of the supplementary material.

---

> > ### Comment · Reviewer_QvG4 · 2023-08-14
> >
> > I would like to thank the authors for addressing my concern. I modified my score accordingly.

---

> > > ### Author Response · Authors · 2023-08-14
> > >
> > > Dear Reviewer QvG4,
> > >
> > > Thank you very much for your kind reply! We will revise our paper to highlight the technical contributions.
> > >
> > > Best
> > > Authors

---

### Official Review · Reviewer_WjRx · 2023-07-04

**Soundness:** 4 excellent
**Presentation:** 3 good
**Contribution:** 3 good
**Rating:** 7
**Confidence:** 3

**Summary:**

This paper proposes two novel algorithms which improve computational complexity and regret bounds over previous algorithms for generalized linear bandits with heavy-tailed rewards by combining the truncation strategy (or means of medians strategy) with the online Newton step.

**Strengths:**

This work improves the computational complexity and regrets bound of the generalized linear contextual bandits with heavy-tailed rewards which are closely related to the real-world application.

**Weaknesses:**

1. Among the input values for the algorithm, $S$, $\epsilon$ and $v$ is not known in practice.


**Questions:**

1. Can dependency on $\kappa$ in the regret bound be relaxed?

**Limitations:**

Yes. This work is mostly theoretical and potential negative societal impact is unseen.

---

> ### Author Rebuttal · Authors · 2023-08-07
>
> Thank you sincerely for your time and effort in reviewing our work. We have carefully considered your concerns and our responses are provided as follows.
>
> ----
> **Weeknesses: Among the input values for the algorithm, $S, \epsilon$ and $v$ is not known in practice.**
>
> Huang et al. [2022] proposed an algorithm for heavy-tailed MAB models that eliminates the dependence on $\epsilon$ and $v$ by utilizing the FTRL framework and the doubling trick technique. This provides some insights to remove the dependence on $\epsilon$ and $v$ in the GLB model. However, GLB algorithm is based on the online Newton step method, which is fundamentally different from FTRL, and we will strive to address this issue in future research. For the parameter $S$, most existing online GLB algorithms requires prior knowledge $S$ [Zhang et al., 2016; Jun et al., 2017; Lu et al., 2019]. We will try to remove this limitation in the future.
>
> -----
> **Questions: Can dependency on $\kappa$ in the regret bound be relaxed?**
>
> [1] introduced an algorithm for the Logistic model that enhanced the dependency on $\kappa$ from $O(\kappa^{-1})$ to $O(\kappa^{-1/2})$. However, their algorithm is offline, as it relies on the maximum likelihood estimate. Consequently, directly applying [1]'s technique to our algorithms would not yield a similar improvement, since our algorithms are online. In future research, we aim to improve the dependency on $\kappa$ by employing a variant of [1]'s technique. The revised version of our paper will provide a more comprehensive discussion of this issue.
>
>
> References
>
> [1] Improved Optimistic Algorithms for Logistic Bandits. Louis Faury, Marc Abeille, Clément Calauzènes, Olivier Fercoq, 2020.

---

> > ### Comment · Reviewer_WjRx · 2023-08-18
> >
> > I thank the authors for their helpful responses. My questions are resolved.

---

### Decision · Program_Chairs · 2023-09-21

**Decision:**

Accept (poster)

**Comment:**

All reviewers have reached a consensus and recommend to accept the paper. The main strengths are:
* clarity of the algorithm and presentation
* relevant problem addressed.

The discussion focused on the potential weakness of the technical contribution that the authors eventually addressed in their rebuttal. We recommend that the final version reflects these exchanges and highlight the novel technical contributions precisely.